# SIMPLICIAL EMBEDDINGS IN SELF-SUPERVISED LEARNING AND DOWNSTREAM CLASSIFICATION

**Samuel Lavoie**◇∗, **Christos Tsirigotis**◇, **Max Schwarzer**◇, **Ankit Vani**◇,
**Michael Noukhovitch**◇, **Kenji Kawaguchi**‡, **Aaron Courville**◇♣
◇ Mila, Université de Montréal, ‡ National University of Singapore, ♣ CIFAR Fellow

## ABSTRACT

Simplicial Embeddings (SEM) are representations learned through self-supervised learning (SSL), wherein a representation is projected into $L$ simplices of $V$ dimensions each using a softmax operation. This procedure conditions the representation onto a constrained space during pre-training and imparts an inductive bias for discrete representations. For downstream classification, we provide an upper bound and argue that using SEM leads to a better expected error than the unnormalized representation. Furthermore, we empirically demonstrate that SSL methods trained with SEMs have improved generalization on natural image datasets such as CIFAR-100 and ImageNet. Finally, when used in a downstream classification task, we show that SEM features exhibit emergent semantic coherence where small groups of learned features are distinctly predictive of semantically-relevant classes.

## 1 INTRODUCTION

Self-supervised learning (SSL) is an emerging family of methods that aim to learn representations of data without manual supervision, such as class labels. Recent works (Hjelm et al., 2019; Grill et al., 2020; Saeed et al., 2020; You et al., 2020) learn dense representations that can solve complex tasks by simply fitting a linear model on top of the learned representation. While SSL is already highly effective, we show that changing the *type* of representation learned can improve both the performance and interpretability of these methods.

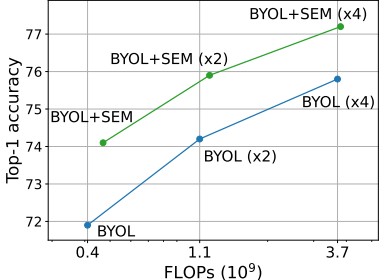

Figure 1: Linear probe accuracy of BYOL and BYOL + SEM on ImageNet trained for *200 epochs* with a ResNet-50 architecture.

For this we draw inspiration from overcomplete representations: representations of an input that are non-unique combinations of a number of basis vectors greater than the input's dimensionality (Lewicki & Sejnowski, 2000). Mostly studied in the context of the sparse coding literature (Gregor & LeCun, 2010; Goodfellow et al., 2012; Olshausen, 2013), sparse overcomplete representations have been shown to increase stability in the presence of noise (Donoho et al., 2006), have applications in neuroscience (Olshausen & Field, 1996; Lee et al., 2007), and lead to more interpretable representations (Murphy et al., 2012; Fyshe et al., 2015; Faruqui et al., 2015). But, the basis vector is learned using linear models (Lewicki & Sejnowski, 2000; Teh et al., 2003).

In this work, we show that SSL may be used to learn discrete, sparse and overcomplete representations. Prior work has considered sparse representation but not sparse and overcomplete representation learning with SSL; for example, Dessì et al. (2021) propose to discretize the output of the encoder in a SSL model using Gumbel-Softmax (Jang et al., 2017). However, we show that discretization during pre-training is not necessary to achieve a sparse representation. Instead, we propose to project the encoder's output into $L$ vectors of $V$ dimensions onto which we apply a softmax function to impart an inductive bias toward sparse one-hot vectors (Correia et al., 2019; Goyal et al., 2022), also alleviating the need to use high-variance gradient estimators to train the encoder. We refer to this embedding as Simplicial Embeddings (SEM), as the softmax functions map the unnormalized representations onto $L$ simplices. The procedure to induce SEM is simple, efficient, and generally applicable.

---

∗Correspondence to: samuel.lavoie.m@gmail.com

The SSL pre-training phase, used with SEM, learns a set of $L$ *approximately* one-hot vectors. Key to controlling the inductive bias of SEM during pre-training is the softmax temperature parameter: the lower the temperature, the stronger the bias toward sparsity. Consistent with earlier attempts at sparse representation learning (Coates & Ng, 2011), we find that the optimal sparsity for pre-training need not match the optimal level for downstream learning.

For downstream classification, we may discretize the learned representation by, for example, taking the argmax for each simplex. But, we can also use SEM to control the representation's expressivity via the softmax's temperature. We provide a theoretical bound showing that the expected error follows a trade-off between the training error and the representations' expressivity that can be controlled by the softmax's temperature used to normalize the representation for downstream classification. Our bound also shows improved expected error as we increase $L$ and $V$ for SEM.

SEM is generally applicable to recent SSL methods. Applying it to seven different SSL methods (Chen et al., 2020b; He et al., 2020; Grill et al., 2020; Caron et al., 2020; 2021; Zbontar et al., 2021; Bardes et al., 2022), we find accuracy increases of 2% to 4% on CIFAR-100. We observe monotonic improvement as we increase the number of vectors $L$, showing the benefit of the overcomplete representations learned by SEM, while this improvement is absent when we do not use softmax normalization. When training a SSL method with SEM on ImageNet we also observe improvements on in-distribution compared to the baseline (Figure 1). We also observe improvement on out-of-distribution test sets, semi-supervised learning benchmark and transfer learning datasets, demonstrating the potential of SEM for large scale applications. Finally, we find that SEM learns features that are closely aligned to the semantic categories in the data. This demonstrates that SEM learns disentangled and interpretable representations, as previously observed in overcomplete representations (Faruqui et al., 2015).

## 2 RELATED WORK

The softmax operation has been used in other contexts, notably as an architectural component for models to attend to context-dependent queries via, for example, an attention mechanism (Bahdanau et al., 2016; Vaswani et al., 2017; Correia et al., 2019; Goyal et al., 2022), a mixture of experts (Jordan & Jacobs, 1993) or memory augmented networks (Graves et al., 2014). This operation is also used for the computation of several SSL objectives such as InfoNCE (van den Oord et al., 2018; Hjelm et al., 2019), and as a normalization of the output to compute the objective in DINO and SWaV (Caron et al., 2020; 2021). Different from these, our method places the softmax at the output of an encoder to constrain the representation into a set of $L$ sparse vectors.

Similar to our approach, other architectural constraints such as Dropout (Srivastava et al., 2014), BatchNorm (Ioffe & Szegedy, 2015) and LayerNorm (Ba et al., 2016) also improve the training of large neural networks. However, contrary to SEMs, they are not used to induce sparsity on the representation or control its expressivity for downstream tasks. Closer to our work, Liu et al. (2021) propose to constrain the expressivity of the representation of a neural network with a set of discrete-valued symbols obtained using a set of Vector Quantized (Oord et al., 2018) bottlenecks. Similarly, Dessi et al. (2021) propose a communication game with a discrete bottleneck. The idea of discretizing the encoder's output is similar to using SEM vectors that are one-hot (e.g. temperature $= 0$) and only one symbol (e.g. $L = 1, V = 2048$). In our work, we find success in removing the hard-discretization and having $L > 1$, which can be interepreted as combining several symbols.

## 3 SIMPLICIAL EMBEDDINGS

Simplicial Embeddings (SEM) are representations that can be integrated easily into a contrastive learning model (Hjelm et al., 2019; Chen et al., 2020b), the BYOL method (Grill et al., 2020), and other SSL methods (Caron et al., 2020; 2021; Zbontar et al., 2021). For example, in BYOL, we insert the SEM after the encoder and before the projector and the rest is unchanged as shown in Figure 2c. In this figure, $t$ and $t'$ are augmentations defined by the practitioner, $\xi$ are parameters of the target network that are updated as moving average of the parameters $\theta$ of the online networks trained with SGD. So, $\xi$ are updated as follow: $\xi \leftarrow \alpha\xi + (1-\alpha)\theta$, with $\alpha \in [0,1]$.

To produce SEM representation, the encoder's output $e$ is embedded into $L$ vectors $z_i \in \mathbb{R}^V$. A temperature parameter $\tau$ scales $z_i$, and then a softmax re-normalizes each vector $z_i$ to produce $\bar{z}_i$.

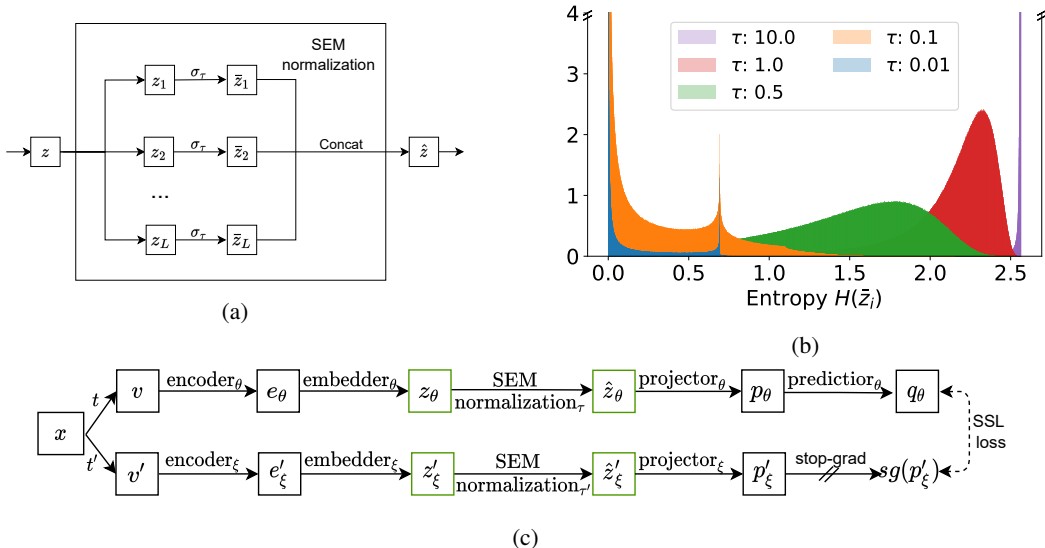

Figure 2: **(a)** Procedure to obtain Simplicial Embeddings (SEM). A matrix $z \in \mathbb{R}^{L \times V}$ contains $L$ vectors $z_i \in \mathbb{R}^V$. The vectors $z_i$ are normalized with $\sigma_\tau$, the softmax operation with temperature $\tau$. The normalized vectors are concatenated into the vector $\hat{z}$. **(b)** Normalized histogram of the entropies $H(\bar{z}_i)$ of each simplex $\bar{z}_i$ for the sample in CIFAR's training dataset at the end of pre-training with various $\tau$. The peak at $\ln(2)$ for $\tau = 0.01$ and $\tau = 0.1$ are a large number of simplices with two elements close to 0.5. **(c)** Integration of SEM with BYOL (Grill et al., 2020). The encoder outputs a latent vector which is embedded into the matrix $z \in \mathbb{R}^{L \times V}$ and then transformed into SEM.

Finally, the normalized vectors $\bar{z}_i$ are concatenated to produce the vector $\hat{z}$ of length $L \cdot V$. We illustrate SEM in Figure 2a. Formally, the re-normalization is as follows:

$$\bar{z}_i := \sigma_\tau(z_i), \quad \sigma_\tau(z_i)_j = \frac{e^{z_{ij}/\tau}}{\sum_{k=1}^{V} e^{z_{ik}/\tau}}, \quad \hat{z} := \text{Concat}(\bar{z}_1, \dots, \bar{z}_L), \quad \forall i \in [L], \forall j \in [V]. \quad (1)$$

## 3.1 INDUCTIVE BIAS TOWARDS SPARSITY DURING PRE-TRAINING

In SEM, $L$ controls the numbers of simplices and $V$ controls the dimensionality of each simplex. As such, the higher $V$ is, the sparser the representation can be. During pre-training, the constraint induced by embedding the representation into a simplex biases each vector towards sparse vectors by creating a zero-sum competition between the components of the vector. In order for a component to increase by $\alpha$, then the other elements must decrease by $\alpha$, and all elements are bounded by 0. For networks to learn useful features and minimize their objective, they must prioritize some components at the expense of others. The strength of this bias is controlled via the pretraining temperature $\tau_p$ of the softmax, and the size of the vectors $V$ as it was noted in the context of attention (Vaswani et al., 2017; Wang et al., 2021b). For SSL methods with a target network, the temperature for the target network can be different to the online network's as no gradient is back-propagated through it.

To visualize the effect of the temperature on SEM after pre-training, we interpret each simplex as a probability mass function $p(\bar{z}_{ij})$ where, for all $i \in [L]$, $\sum_{j=1}^{V} p(\bar{z}_{ij}) = 1$ and $p(\bar{z}_{ij}) \geq 0 \; \forall j$. The entropy of a simplex $\bar{z}_i$, defined as $H(\bar{z}_i) := -\sum_{j=1}^{V} p(\bar{z}_{ij}) \log p(\bar{z}_{ij})$, informs whether the simplex is a sparse or a dense vector. That is, if $H(\bar{z}_i^{(x)}) = 0$ then the vector is one-hot. On the other hand, if $H(\bar{z}_i^{(x)}) = \ln(V)$ then the vector is dense and uniform. While the temperature $\tau_p$ is merely a scaling of the logits, it has an important control over the learned representation's entropy and resulting SEM sparsity. We demonstrate this by learning a representation on CIFAR-100 using BYOL, and analyze the entropies of the resulting simplices. In Figure 2b, we plot the histogram of the entropies $H(\bar{z}_i)$, for a given $\tau_p$, of each simplex for each sample in the training set of CIFAR-100. We observe that

even after pre-training, small temperatures ($\tau_p = 0.01$) yields representations that are close to one-hot vectors while high temperatures yields vectors that are close to uniform vectors.

By pre-training using a softmax, SEMs create representations that are conditioned to fit onto simplices. In pre-training, we select $\tau_p$ for optimal inductive bias: $\tau_p$ too small yields vanishing gradients (Wang et al., 2021b) and $\tau_p$ too large yields a bias that is too weak. We may select a different optimal $\tau_d$ for downstream performance as discussed formally in the next subsection.

## 3.2 SEM IMPROVEMENT ON THE GENERALIZATION OF THE DOWNSTREAM CLASSIFIER

In this subsection, we theoretically demonstrate the benefit of training a downstream classifier with SEM normalized input compared to a baseline classifier with unnormalized input. We show that: (1) there is a trade-off between the training loss and the generalization gap, which is controlled by the value of $\tau_d$ (denoted $\tau := \tau_d$ in this subsection), (2) SEM can improve the base model performance when we attain good balance in this trade-off, and (3) the improvement due to SEM is expected to increase or stay constant as $L$ and $V$ increase. In the remainder of this subsection, we introduce the notation and assumptions needed to understand and derive the result, then present our theoretical claim and discuss its implications.

**Notation.** We use a training dataset $S = (z^{(i)}, y^{(i)})_{i=1}^n$ of $n$ samples for supervised training of a classifier, using the representation $z$ extracted from the pre-trained model[*] and the corresponding label $y \in \mathcal{Y}$ where $\mathcal{Y}$ is the space of possible labels. Assume that $z \in \mathcal{Z} = [-1, +1]^{L \times V}$, which means that $z$ is a matrix with $L$ rows and $V$ columns. We denote the element of $z$ at row $i$ and column $j$ as $z_{ij}$. Let $g$ represent the downstream classifier. We refer to the baseline downstream model with unnormalized input as $f_{\text{base}}$, and $f_{\text{base}}(z) = g(z)$. The corresponding downstream model trained with the SEM normalization is $f_{\text{SEM}(\tau)}(z) = (g \circ \sigma_\tau)(z)$, where $\sigma_\tau$ is applied element-wise along each row of $z$ such that $\sigma_\tau(z_{ij}) = \frac{e^{z_{ij}/\tau}}{\sum_{t=1}^V e^{z_{it}/\tau}}$ for $j = 1, \ldots, V$. Moreover, we define $f_{\text{base}}^S$ and $f_{\text{SEM}(\tau)}^S$ the base and the SEM normalized models obtained by fitting the dataset $S$. Finally, let $\mathcal{H}$ be the union of the hypothesis spaces of $f_{\text{SEM}(\tau)}$ and $f_{\text{base}}$.

To compare the quality of the base model and the model with SEM normalization, we analyze the generalization gap $\mathbb{E}_{z,y}[l(f_S(z), y)] - \frac{1}{n} \sum_{i=1}^n l(f_S(z^{(i)}), y^{(i)})$ for each $f_S \in \{f_{\text{SEM}(\tau)}^S, f_{\text{base}}^S\}$, where $l : \mathbb{R} \times \mathcal{Y} \to \mathbb{R}_{\geq 0}$ is the per-sample loss.

The key insight that we exploit for the theorem is that the softmax operation $\sigma_\tau$ controls the expressivity of the input's representation to $g$ via the temperature $\tau$. We denote $\varphi_{f_{\text{base}}}$ as an upper bound on the expressivity of $z_i$ for the baseline model $f_{\text{base}}$, and $\varphi_{f_{\text{SEM}(\tau)}}$ as the upper bound on the expressivity of $\sigma_\tau(z_i)$ for the model with SEM normalization $f_{\text{SEM}(\tau)}$. The formal definition of $\varphi_{f_{\text{base}}}$ and $\varphi_{f_{\text{SEM}(\tau)}}$ requires proof devices that will hinder the readability of this section, so we refer the reader to Appendix A for a detailed definition. Let $\varphi_f \in \{\varphi_{f_{\text{base}}}, \varphi_{f_{\text{SEM}(\tau)}}\}$. Intuitively, $\varphi_{f_S}$ measures the largest possible distance that two embeddings can have such that the largest component remains the same for both embeddings. We note that this measure depends only on $V$ for $f_{\text{base}}$, and on both $V$ and $\tau$ for $f_{\text{SEM}(\tau)}$. We use $\varphi_{f_S}(V, \tau)$ to denote the measure given by either model and note that $\tau$ has no effect for $f_{\text{base}}$.

**Assumptions.** We assume that the per-sample loss is bounded such that $l(f(z), y) \leq B$ for all $f \in \mathcal{H}$ and for all $(z, y) \in \mathcal{Z} \times \mathcal{Y}$. For example, $B = 1$ for the 0-1 loss. Next, let $l_y$ be the per-sample loss given $y$. We assume that $l_y \circ g$ are uniformly Lipschitz functions for all $y \in \mathcal{Y}$ and $g \in \mathcal{G}_S$, where $\mathcal{G}_S$ is the set of classifiers $g$ returned by the training algorithm using the dataset $S$. Let $R$ be such a uniform Lipschitz constant. This means that $|(l_y \circ g)(\sigma_f(z)) - (l_y \circ g)(\sigma_f(z'))| \leq R\|\sigma_f(z) - \sigma_f(z')\|_F$, where $l_y(g \circ \sigma_f(z)) = l(g \circ \sigma_f(z), y)$, and $\sigma_f = \sigma_\tau$ when $f = f_{\text{SEM}(\tau)}$ and $\sigma_f$ is identity when $f = f_{\text{base}}$. Finally, we assume that there exists $\Delta > 0$ such that for all representations $z$ of the underlying distribution we have that for any $i \in [L]$, if $k = \arg\max_{j \in [V]} z_{ij}$, then $z_{ik} \geq z_{ij} + \Delta$ for any $j \neq k$. Since $\Delta$ can be arbitrarily small (e.g. as small as machine precision), this assumption typically holds in practice. We are now ready to state our theoretical claim.

Theorem 1 illuminates the advantage of SEM and the effect of the hyper-parameter $\tau$ on the performance of the downstream classifier. We present the proof in Appendix A.

---

[*]In this subsection, we refer to the extracted representation as $z$, the embedder's output

**Theorem 1.** *Let $V \geq 2$. For any $1 \geq \delta > 0$, with probability at least $1 - \delta$, the following holds for any $f_S \in \{f_{\text{SEM}(\tau)}^S, f_{\text{base}}^S\}$:*

$$\mathbb{E}_{z,y}[l(f_S(z), y)] \leq \frac{1}{n} \sum_{i=1}^{n} l(f_S(z^{(i)}), y^{(i)}) + R\sqrt{L\,\varphi_{f_S}(V, \tau)} + c\sqrt{\frac{\ln(2/\delta)}{n}},$$

*where $c > 0$ is a constant in $(n, f, \mathcal{H}, \delta, \tau, S)$. Moreover,*

$$\varphi_{f_{\text{SEM}(\tau)}^S} \to 0 \quad as \quad \tau \to 0 \quad and \quad \varphi_{f_{\text{SEM}(\tau)}^S} - \varphi_{f_{\text{base}}^S} \leq \frac{3}{4}(1 - V) < 0 \quad \forall \tau > 0.$$

The first statement of Theorem 1 shows that the expected loss is bounded by the three terms: the training loss $\frac{1}{n} \sum_{i=1}^{n} l(f_S(z^{(i)}), y^{(i)})$, the second term $R\sqrt{L\varphi_{f_S}}$, and the third term $c\sqrt{\frac{\ln(2/\delta)}{n}}$. Since $c$ is a constant in $(n, f, \mathcal{H}, \delta, \tau, S)$, the third term goes to zero as $n \to \infty$ and is the same with and without SEM. Thus, for the purpose of assessing the impact of SEM, we can focus on the second term, where a difference arises. Theorem 1 shows that $R\sqrt{L\varphi_{f_S}}$ goes to zero with SEM; i.e., $\varphi(f_{\text{SEM}(\tau)}^S) \to 0$ as $\tau \to 0$. Also, for any $\tau > 0$, the second term with SEM is strictly smaller than that without SEM as $\varphi_{f_{\text{SEM}(\tau)}^S} - \varphi_{f_{\text{base}}^S} \leq \frac{3}{4}(1 - V) < 0$ and demonstrates that the improvement due to SEM is expected to asymptotically increase as $V$ increases. Moreover, $L$ is a multiplicative constant of $\varphi$ which shows that, as $L$ increases, the improvement due to SEM is also expected to be higher. Overall, Theorem 1 shows the benefit of SEM as well as the trade-off with $\tau$. When $\tau \to 0$, the second term goes to zero, but the training loss (the first term) can increase due to underfitting resulting from the reduction in representation expressivity. Thus, $\tau$ should be chosen to optimally balance this trade-off.

## 4 EMPIRICAL ANALYSIS

We empirically study the effect of SEM on the representation of SSL methods and demonstrate that SEM improves the test set accuracy on CIFAR-100 (Krizhevsky, 2009). We compare SEM with other methods for inducing sparse representations during pretraining and demonstrate that SEM lead to better downstream accuracy. On IMAGENET (Deng et al., 2009), we study the effect of SEM on robustness, semi-supervised learning and transfer learning datasets, demonstrating consistent improvement attributed to SEM. Finally, we present evidences that features produced by SEMs are more naturally aligned with the semantic categories of the data. The code for reproducing the results is available at: https://github.com/lavoiems/simplicial-embeddings/.

**Training setup.** For all experiments, we build off the implementation of the baseline models from the Solo-Learn library (da Costa et al., 2021). We probe the encoder's output for the baseline methods, as typically done in the literature. For models with SEM, we probe the SEM normalized representation (i.e. $\hat{z}$). In our experiments, the embedder is a linear layer followed by BatchNorm (Ioffe & Szegedy, 2015). Unless mentioned otherwise, we use $L = 5000$ and $V = 13$ for the SEM representation. We do not perform any search for the non-SEM hyper-parameters. The SEM hyper-parameters are selected by using a validation set of 10% of the training set of CIFAR-100 and 10 samples per class on the in distribution dataset for IMAGENET. The test accuracy is obtained by retraining the model with all of the training data using the parameters found with the validation set. We pre-train the SSL models for 200 epochs on IMAGENET and 1000 epochs on CIFAR-100.

Table 1: Linear probe top-1 accuracy on CIFAR-100 trained for *1000 epochs* with a ResNet-18/50 encoder. We compare the *test accuracy* of several SSL models with and without SEM. **Boldface** indicates highest accuracy. Green rows indicate a SSL method + SEM.

|  | SIMCLR | MOCO | BYOL | BARLOW-TWINS | SWAV | DINO | VICREG |
|---|---|---|---|---|---|---|---|
| *ResNet-18:* | | | | | | | |
| Baseline | 66.8 | 69.3 | 70.7 | 70.7 | 64.6 | 66.8 | 68.5 |
| With SEM | **69.5** | **71.0** | **73.9** | **73.0** | **67.7** | **69.2** | **71.4** |
| *ResNet-50:* | | | | | | | |
| Baseline | 70.5 | 73.24 | 74.2 | 72.0 | – | – | 70.8 |
| With SEM | **73.2** | **75.8** | **77.5** | **73.3** | – | – | **73.3** |

### 4.1 SEM IMPROVES ON DOWNSTREAM CLASSIFICATION

**Baseline comparison.** We evaluate the effect of adding SEMs in seven modern SSL approaches. We take standard SimCLR (Chen et al., 2020b), MoCo-v2 (He et al., 2020), BYOL (Grill et al., 2020) Barlow-Twins (Zbontar et al., 2021), SwAV (Caron et al., 2020), DINO (Caron et al., 2021) and VicReg (Bardes et al., 2022) models and implement SEM after the encoder. We compare our approach on CIFAR-100 with a ResNet-18 and ResNet-50 in Table 1. We found SWaV and DINO to be unstable with ResNet-50 thus have decided not to compare them with SEM. For every SSL methods, using SEMs improves the baseline methods by $2\%$ to $4\%$ demonstrating that SEM is a general approach that improves in-distribution generalization for SSL methods.

**Increasing the representation's size of SEM increases the performance.** We find that increasing $L$ (the number of simplices of SEM) beyond the over-complete regime increases the downstream accuracy. This increased performance is not observed when we abstain from using the softmax normalization of SEM. In Figure 3, using a ResNet-50 encoder, we compare BYOL + SEM, with an identical model without the Softmax normalization which we call BYOL + Embed and BYOL to which we increase the representation's size before the mean-pooling using the method proposed in (Dubois et al., 2022) and described in their Appendix F. To be clear, the extracted representation of BYOL + Embed is the embedder's output $z_\theta$ and the extracted representation for BYOL is the encoder's output $e_\theta$. We fix $V = 13$ and scale $L \in [10, 10000]$ to get a range of representation sizes.

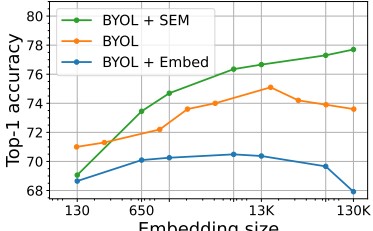

Figure 3: Effect of the Softmax when scaling up $L$ on the linear probe accuracy. Using a RN-50.

Table 2: Comparing SEM with *hard* discretization using Gumbel Softmax (G.S.) and Vector Quantization (V.Q.). RN-18 base on CIFAR-100.

| Accuracy | $e_\theta$ | $\hat{z}_\theta$ |
|---|---|---|
| BYOL | 70.7 | - |
| BYOL+G.S. | 63.3 | 54.5 |
| BYOL+V.Q. | 65.6 | 60.3 |
| BYOL+SEM ($\tau_\mathrm{d} = 0$) | - | **73.2** |
| BYOL+SEM ($\tau_\mathrm{d} = 0.1$) | - | **73.9** |

**Comparison of SEM with *hard* discretization approaches.** Several other methods can be used to induce a sparse and over-complete representation during pre-training and downstream classification. For example, we may sample $L$ discrete one-hot codes of $V$ dimensions using Gumbel Softmax (Jang et al., 2017) as done in Dessì et al. (2021). We can also use Vector Quantization (VQ) (Oord et al., 2018) and consider $L$ latent embedding spaces with $V$ embedding vectors each, wherein the vectors are in $\mathbb{R}^d$. In contrast to SEM, it is not possible to propagate the gradient through the bottleneck trivially and VQ uses straight-through estimation in the embedding space to back-propagate the gradient to the encoder. Here, we observe that these alternative approaches exhibit a considerable decrease in performance in comparison to the baseline as demonstrated in Table 2. In this table, we reproduce the same setup as SEM but we replace the Softmax with hard discretization baselines methods. For discretization with Gumbel Straight-Through estimation, we use the same setup as SEM with $L = 5000$ and $V = 13$, that is 5000 one-hot vectors of 13 dimensions and $\tau = 2$[†]. For VQ, we found that $L = 512$ and $V = 128$ led to the best performance. That is, we have 512 latent embedding spaces, each with 128 possible embedding vectors that are in $\mathbb{R}^{32}$.

We note that while we have not found hard-discretization to be successful during pre-training, we may hard-discretize a SEM representation for downstream task. In Table 2, we also present SEM with $\tau_\mathrm{DS} = 0$, which correspond to using the discretized representation for downstream classification. We obtain the discrete representation by taking the argmax for each simplex. This result demonstrating that SEM with pre-training can be used to learn meaningful discrete codes for downstream applications and yields better performance than the baselines, implying that pre-training with SEM could be be used in applications that require discretization.

**Memory and computational efficiency of SEM.** SEM's performance improvements come at a cost of increased memory allocation (VRAM) due to additional parameters needed to perform the matrix multiplication, and slightly more computation (FLOPs/sample). For very large over-complete representation the increased memory requirement can impede practical application. We propose a more efficient version of SEM by sparsifying the matrix multiplication of the embedder and of the projector and detail this procedure in Appendix D.1. As shown in Table 17, SEM with sparse matrix

---

[†]A hyper-parameter search was performed to select the best performing hyper-parameter.

multiplication use only slightly more memory and compute but outperforms the BYOL baseline on CIFAR-100 though underperforming the regular SEM. We also note that SEM's memory cost becomes relatively minor as we scale up the encoder. As well, the computational cost of SEM is small compared to the total cost of pre-training and achieves higher accuracy using fewer FLOPs compared to scaling the encoder as shown in Figure 1.

## 4.2 ANALYZING THE PARAMETERS OF SEM

We present two figures in this section to better understand the effect of the parameters of SEM on the downstream accuracy. In Figure 4, we evaluate the effect of changing $\tau_p$ and $\tau_d$ on the downstream accuracy. In Figure 5, we evaluate the effect of $L$ and $V$ on the downstream accuracy and also contrast $f_{\text{base}}$ and $f_{\text{SEM}}(\tau = 1)$ by using the same encoder pre-trained with SEM. This allows us to relate some observations to the theorey presented in Section 3.2. Now, we discuss the effect of each of SEM's parameter on the resulting downstream classification.

**Increasing $V$ yields a steep performance increase for small $V$ but quickly plateau.** In Figure 5b, we observe a steep increase of the accuracy for $V < 13$ followed by a plateau for $V > 13$. In Figure 4a, we observe that the optimal accuracy obtained for $V = 1024$ and $L = 64$ is similar to the one obtained for $L = 50$ (Embedding size=650) in Figure 3.

**Increasing $L$ yields monotonical improvement for downstream classification.** In the regime that we can test it, increasing $L$ lead to consistent improvement on the downstream accuracy as observed in Figure 3 and Figure 5a. Using SEM in pre-training only is not enough and using it in the downstream classifier is necessary for the improved performance as demonstrated in Figure 5a.

**The optimal $\tau_p$ depends on $V$.** As previously noted in the context of Attention (Vaswani et al., 2017; Wang et al., 2021a), the optimal attention's temperature is proportional to attention's vector size. We also observe this in SEM. As presented in Figure 4a, the optimal $\tau_p$ for larger $V$ is higher.

**Models with larger $L$ are more robust to smaller $\tau_d$.** In Figure 4, we observe that SSL models are more robust to smaller $\tau_d$ as $L$ increase. We speculate that the information can be scattered across the simplices for large $L$, allowing to reduce the expressivity of each vector with minimal impact on the downstream accuracy.

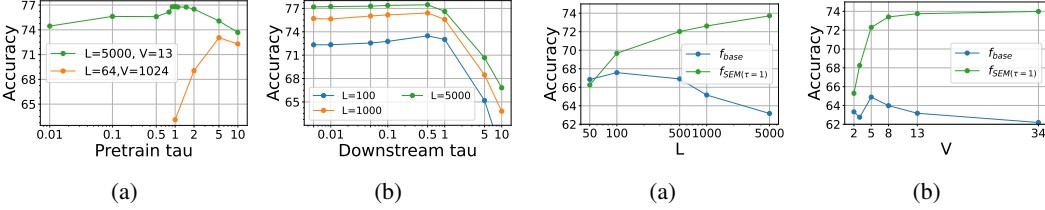

Figure 4: Effect of $\tau_p$ and $\tau_d$ on a RN-50.    Figure 5: Comparing $f_{\text{SEM}}$ and $f_{\text{base}}$ on a RN-18.

## 4.3 SEM IMPROVEMENT ON LARGE-SCALE DATASETS WITH IMAGENET

Figure 1 in the introduction demonstrates that using SEM leads to better in distribution generalization for IMAGENET and is a more efficient method of scaling up the model as compared to scaling up the width of the ResNet-50 encoder. Here, we demonstrate that SEM generally improves the accuracy on several robustness test sets, a semi-supervised learning benchmark and transfer learning datasets. We use BYOL+SEM with an embedding size of 105 000 features ($L = 5000$ and $V = 21$) for these experiments. The embedding is pre-trained for 200 epochs using the BYOL SSL procedure.

**Robustness to out-of-distribution test sets.** We perform a comparative study using several test sets: (**IN**) the in-distribution test set provided in IMAGENET; (**IN-C**) IMAGENET-C, which exhibits a set of common image corruptions (Hendrycks & Dietterich, 2019); (**IN-R**) IMAGENET-R (Hendrycks et al., 2021) which consists of different renderings for several IMAGENET classes; and (**IN-V2**) IMAGENET-V2 (Recht et al., 2019), a distinct test set for IMAGENET collected using the same process; (**IN-A**) IMAGENET-A (Chen et al., 2020a) contains a set of samples that are miclassifier by a IMAGENET ResNet-50 classifier. We use the methodology and software proposed in Djolonga et al. (2020; 2021)

Table 3: Robustness via linear probe top-1 test accuracies on IMAGENET variant datasets, using representations pre-trained for *200 epochs*. * Taken from (Chen & He, 2020)

|  | IN | IN-V2 | IN-R | IN-C | IN-A |
|---|---|---|---|---|---|
| BYOL* | 70.6 | - | - | - | - |
| BYOL | 71.9 | 59.2 | 18.8 | 39.5 | 1.65 |
| BYOL+SEM | **74.1** | **61.2** | **22.1** | **43.4** | **2.53** |

Table 4: Top-1 transfer learning accuracy from IMAGENET pre-trained representation.

|  | FOOD101 | C10 | C100 | SUN | DTD | FLOWER |
|---|---|---|---|---|---|---|
| *Linear probe:* |  |  |  |  |  |  |
| BYOL | 74.2 | 91.8 | 74.9 | 60.9 | **72.2** | 88.9 |
| BYOL+SEM | **74.7** | **93.5** | **78.6** | **62.1** | 71.9 | **91.5** |
| *Fine-tuned:* |  |  |  |  |  |  |
| BYOL | 83.1 | 97.2 | 83.6 | 59.1 | 69.2 | 85.4 |
| BYOL+SEM | **84.7** | 97.2 | **85.6** | **63.3** | **71.3** | **91.7** |

to perform our experiments. We observe that BYOL + SEM outperforms BYOL on every robustness datasets probed, demonstrating that SEM also improves generalization to out-of-distribution test sets.

**Transfer learning.** We probe the effectiveness of SEM in BYOL and MoCo when transferring representations trained on IMAGENET to other classification tasks. We follow the linear evaluation and fine-tuning methodologies described in previous works (Grill et al., 2020; Lee et al., 2021), which entails training a linear classifier with logistic regression using sklearn (Pedregosa et al., 2011) on the embeddings of the samples and fine-tuning the encoder respectively. To avoid out-of-memory issues that may occur in the linear probe experiment with the sklearn solver when the number of features, we discretize our features and use sparse matrix to fit the logistic regression. This is equivalent to forcing $\tau_d = 0$ for all the experiments. For the fine-tuning experiments, we fix $\tau_d = 1$ since the evaluation method allows for mini-batch gradient descent. We perform our transfer learning experiments on the following datasets: FOOD (Bossard et al., 2014), CIFAR-10 (C-10) (Krizhevsky, 2009), CIFAR-100 (C-100) (Krizhevsky, 2009), SUN (Xiao et al., 2010), DTD (Cimpoi et al., 2014) and FLOWER (Nilsback & Zisserman, 2008).

This task evaluates the generality of the encoder as it has to encode samples from various out-of-distribution domains with categories that it may not have seen during training. We present our results in Table 4 and observe that SEM improves the transfer accuracy over the baseline for every datasets but DTD for the linear probe experiment. For DTD, we hypothesize that the drop in performance is due to the fact that we use a temperature that is too small. Since this is a texture dataset with higher frequency, it might be the case that we need more expressivity to correctly fit the data. We support the conjecture with the fine-tuning experiment where BYOL + SEM out-performs the baseline.

**Semi-supervised learning.** We evaluate the effect of using SEM when fine-tuning on a classification task with a small subset of IMAGENET's training set. We follow the semi-supervised learning procedure of Chen et al. (2020b); Grill et al. (2020) and use the same fixed splits of 1% and 10% of ImageNet labelled training set. In Table 5, we demonstrate that using SEM lead to an important increased performance, especially in the low supervised data regime.

Table 5: Semi-supervised learning accuracy by fine-tuning on IMAGENET.

|  | Top-1 | | Top-5 | |
|---|---|---|---|---|
|  | 1% | 10% | 1% | 10% |
| BYOL | 51.6 | 67.5 | 78.0 | 88.9 |
| BYOL+SEM | **56.7** | **69.9** | **81.0** | **90.0** |

## 4.4 SEMANTIC COHERENCE OF SEM FEATURES

Here we demonstrate that SEM features are coherently aligned with the semantics present in the training data. Qualitatively, we visualize the most predictive features of a downstream linear classifier trained on CIFAR-100 and see that the classes with similar predictive features are semantically related. Quantitatively we propose a metric that returns the ratio of features mostly predictive for a classes that are in the same super class to total number of class predictive for this feature.

For both our analysis, we use a linear classifier trained on the features extracted from BYOL with and without SEM. Consider the trained linear classifier with a weight matrix $W \in \mathbb{R}^{N \times C}$, with $N$ features, and $C$ classes. By preserving the top $K$ parameters of the weight matrix $W$ for each class and pruning the features predictive for only one class, we create a bipartite graph between two set of nodes: the CIFAR-100 classes and the features of the representation. We denote this graph $\mathcal{W}_K$.

The qualitative analysis is given by plotting the subset $\mathcal{W}_5$, obtained by taking the top 5 features for each class. We present a subset of the graph for BYOL+SEM in Figure 6a and for BYOL in Figure 6b. The full graphs are presented in the Appendix. In the SEM plot, a set of connected components emerge, and the connected components of the graph are semantically related. For example, the

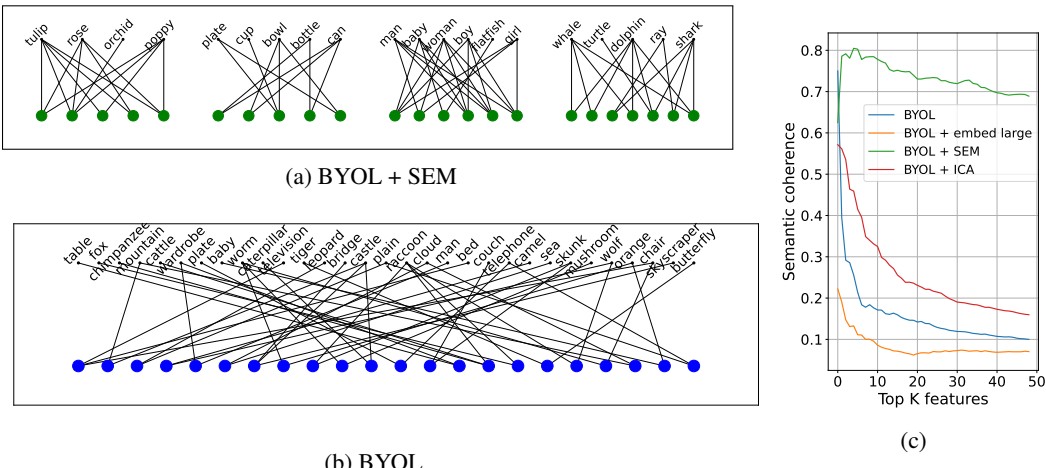

(a) BYOL + SEM

(b) BYOL

(c)

Figure 6: Semantic coherence of the features. **(a)** and **(b)** Subset of $\mathcal{W}_5$, the bipartite graph of the most 5 highest magnitude features on BYOL + SEM features **(a)** and BYOL on the encoded features **(b)**. **(c)** Coherence of the top $K$ features to the semantics of the super-class of the categories of CIFAR-100. It is taken as the number of pairwise categories in the same super-class for which a feature is among its top $K$ most predictive features over the total number of pairwise categories.

first set of connected components are flowers, and the last set of connected components are aquatic mammals.‡. The same class coherence is not observed with either the BYOL baseline or with BYOL augmented with a large representation. In particular, we do not see a small number of semantically related connected components. Instead, we see a large fully connected graphs.

Next, we describe how we quantitatively measure the semantic coherence of the features. Notice that two classes share a common predictive feature on $\mathcal{W}_K$ if they are 2-neighbour. Let $\mathcal{N}(c_i)$ returns all pairs $(c_i, c_j)$ for all $j$ 2-neighbour of $c_i$. Moreover, define the operation $\text{is\_super}(c_i, c_j)$ which returns 1 if $c_i$ and $c_j$ are from the same CIFAR-100 superclass and 0 otherwise. We reproduce the superclass of CIFAR-100 in Table 22 in the Appendix. We measure semantic coherence as follows:

$$\text{Coherence}(\mathcal{W}_K) := \frac{1}{C} \sum_{i=1}^{C} \frac{\sum_{(c_i, c_j) \in \mathcal{N}(c_i)} \text{is\_super}(c_i, c_j)}{|\mathcal{N}(c_i)|}, \tag{2}$$

where $C = 100$ for CIFAR-100 and $|\cdot|$ is the cardinality of a set.

We compare the semantic coherence of BYOL+SEM with the control experiments on BYOL: regular BYOL, BYOL with an embedding of the same size as BYOL+SEM but without the normalization and BYOL to which we applied linear ICA (Hyvärinen & Oja, 2000) in an attempt to disentangle the features. In Figure 10, we plot the full graph $\mathcal{W}_5$ for BYOL+SEM and the baselines. We observe that using the SEM yields semantically coherent features for all the classes of CIFAR-100. This observation is consistent with the qualitative and quantitative experiments presented earlier and demonstrates that SEM's inductive bias during pre-training leads to features that are semantically coherent with the semantic categories extant in the data. This arguably have important implications for improving the interpretability of SSL representations.

## 5 CONCLUSION

SEM is a simple, drop-in module that induces discrete sparse overcomplete representations for standard SSL methods using a softmax operation. This simple modification leads to improved generalization on downstream classification across several state-of-the-art SSL methods. Furthermore, SEM improves performance on out-of-distribution, semi-supervised, and transfer learning tasks across the board and also scales with encoder size. By analyzing semantic coherence, we find that SEMs naturally disentangle data into semantic categories without any explicit training objectives.

---

‡Although "flatfish" may seem out of place in the third set, manually checking CIFAR images showed that many images labelled "flatfish" are often humans holding flatfish.

## ACKNOWLEDGEMENTS

The authors are grateful for the insightful discussions with Xavier Bouthillier, Hattie Zhou, Sébastien Lachapelle, Tristan Deleu, Yuchen Lu, Eeshan Dhekane, Maude Lizaire, Julien Roy and David Dobre. We acknowledge funding support from Samsung and Hitachi, as well as support from Aaron Courville's CIFAR CCAI chair. We also wish to acknowledge Mila and Compute Canada for providing the computing infrastructure that enabled this project. Finally, this project would not have been possible without the contribution of the following open source projects: Pytorch (Paszke et al., 2019), Orion (Bouthillier et al., 2022), Solo-Learn (da Costa et al., 2021), Scikit-Learn (Pedregosa et al., 2011), and Numpy (Harris et al., 2020).

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

# A   Proof of Theorem 1

Let us introduce additional notations used in the proofs. Define $r = (z, y) \in \mathcal{R}$, $\ell(f, r) = l(f(z), y)$,

$$\tilde{\mathcal{C}}_{y,k_1,\ldots,k_L} = \{(z, \hat{y}) \in \mathcal{Z} \times \mathcal{Y} : \hat{y} = y, k_j = \arg\max_{t \in [V]} z_{j,t} \quad \forall j \in [L]\},$$

and

$$\tilde{\mathcal{Z}}_{k_1,\ldots,k_L} = \{z \in \mathcal{Z} : k_j = \arg\max_{t \in [V]} z_{j,t} \quad \forall j \in [L]\}.$$

We then define $\mathcal{C}_k$ to be the flatten version of $\tilde{\mathcal{C}}_{y,k_1,\ldots,k_L}$; i.e., $\{\mathcal{C}_k\}_{k=1}^K = \{\tilde{\mathcal{C}}_{y,k_1,\ldots,k_L,y}\}_{y \in \mathcal{Y}, k_1,\ldots,k_L \in [V]}$ with $C_1 = \tilde{\mathcal{C}}_{1,1,\ldots,1}$, $C_2 = \tilde{\mathcal{C}}_{2,1,\ldots,1}$, $C_{|\mathcal{Y}|} = \tilde{\mathcal{C}}_{|\mathcal{Y}|,1,\ldots,1}$, $C_{|\mathcal{Y}|+1} = \tilde{\mathcal{C}}_{1,2,1,\ldots,1}$, $C_{2|\mathcal{Y}|} = \tilde{\mathcal{C}}_{|\mathcal{Y}|,2,1,\ldots,1}$, and so on. Similarly, define $\mathcal{Z}_k$ to be the flatten version of $\tilde{\mathcal{Z}}_{k_1,\ldots,k_L}$. We also use $\mathcal{Q}_i = \{q \in [-1, +1]^V : i = \arg\max_{j \in [V]} q_j\}$, $\mathcal{I}_k := \mathcal{I}_k^S := \{i \in [n] : r_i \in \mathcal{C}_k\}$, and $\alpha_k(h) := \mathbb{E}_r[\ell(h, r)|r \in \mathcal{C}_k]$. Moreover, we define $\varphi(f_{\text{base}}^S) = \sup_{i \in [V]} \sup_{q,q' \in Q_i} \|q - q'\|_2^2$, and $\varphi(f_{\text{SEM}(\tau)}^S) = \sup_{i \in [V]} \sup_{q,q' \in Q_i} \|\sigma_\tau(q) - \sigma_\tau(q')\|_2^2$ where $\sigma_\tau(q)_j = \frac{e^{q_j/\tau}}{\sum_{t=1}^V e^{q_t/\tau}}$ for $j = 1, \ldots, V$.

We first decompose the generalization gap into two terms using the following lemma:

**Lemma 1.** *For any $\delta > 0$, with probability at least $1 - \delta$, the following holds for all $h \in \mathcal{H}$:*

$$\mathbb{E}_r[\ell(h, r)] - \frac{1}{n}\sum_{i=1}^n \ell(h, r_i) \leq \frac{1}{n}\sum_{k=1}^K |\mathcal{I}_k|\left(\alpha_k(h) - \frac{1}{|\mathcal{I}_k|}\sum_{i \in \mathcal{I}_k} \ell(h, r_i)\right) + c\sqrt{\frac{\ln(2/\delta)}{n}}.$$

*Proof.* We first write the expected error as the sum of the conditional expected error:

$$\mathbb{E}_r[\ell(h, r)] = \sum_{k=1}^K \mathbb{E}_r[\ell(h, r)|r \in \mathcal{C}_k]\Pr(r \in \mathcal{C}_k) = \sum_{k=1}^K \mathbb{E}_{r_k}[\ell(h, r_k)]\Pr(r \in \mathcal{C}_k),$$

where $r_k$ is the random variable for the conditional with $r \in \mathcal{C}_k$. Using this, we decompose the generalization error into two terms:

$$\mathbb{E}_r[\ell(h, r)] - \frac{1}{n}\sum_{i=1}^n \ell(h, r_i) \tag{3}$$

$$= \sum_{k=1}^K \mathbb{E}_{r_k}[\ell(h, r_k)]\left(\Pr(r \in \mathcal{C}_k) - \frac{|\mathcal{I}_k|}{n}\right) + \left(\sum_{k=1}^K \mathbb{E}_{r_k}[\ell(h, r_k)]\frac{|\mathcal{I}_k|}{n} - \frac{1}{n}\sum_{i=1}^n \ell(h, r_i)\right).$$

The second term in the right-hand side of (3) is further simplified by using

$$\frac{1}{n}\sum_{i=1}^n \ell(h, r_i) = \frac{1}{n}\sum_{k=1}^K \sum_{i \in \mathcal{I}_k} \ell(h, r_i),$$

as

$$\sum_{k=1}^K \mathbb{E}_{r_k}[\ell(h, r_k)]\frac{|\mathcal{I}_k|}{n} - \frac{1}{n}\sum_{i=1}^n \ell(h, r_i) = \frac{1}{n}\sum_{k=1}^K |\mathcal{I}_k|\left(\mathbb{E}_{r_k}[\ell(h, r_k)] - \frac{1}{|\mathcal{I}_k|}\sum_{i \in \mathcal{I}_k} \ell(h, r_i)\right)$$

Substituting these into equation (3) yields

$$\mathbb{E}_r[\ell(h, r)] - \frac{1}{n}\sum_{i=1}^n \ell(h, r_i) \tag{4}$$

$$= \sum_{k=1}^K \mathbb{E}_{r_k}[\ell(h, r_k)]\left(\Pr(r \in \mathcal{C}_k) - \frac{|\mathcal{I}_k|}{n}\right) + \frac{1}{n}\sum_{k=1}^K |\mathcal{I}_k|\left(\mathbb{E}_{r_k}[\ell(h, r_k)] - \frac{1}{|\mathcal{I}_k|}\sum_{i \in \mathcal{I}_k} \ell(h, r_i)\right)$$

$$\leq B\sum_{k=1}^K \left|\Pr(r \in \mathcal{C}_k) - \frac{|\mathcal{I}_k|}{n}\right| + \frac{1}{n}\sum_{k=1}^K |\mathcal{I}_k|\left(\mathbb{E}_{r_k}[\ell(h, r_k)] - \frac{1}{|\mathcal{I}_k|}\sum_{i \in \mathcal{I}_k} \ell(h, r_i)\right)$$

By using the Bretagnolle-Huber-Carol inequality (van der Vaart & Wellner, 1996, A6.6 Proposition), we have that for any $\delta > 0$, with probability at least $1 - \delta$,

$$\sum_{k=1}^{K} \left| \Pr(r \in \mathcal{C}_k) - \frac{|\mathcal{I}_k|}{n} \right| \leq \sqrt{\frac{2K \ln(2/\delta)}{n}}. \tag{5}$$

Here, notice that the term of $\sum_{k=1}^{K} \left| \Pr(r \in \mathcal{C}_k) - \frac{|\mathcal{I}_k|}{n} \right|$ does not depend on $h \in \mathcal{H}$. Moreover, note that for any $(f, h, M)$ such that $M > 0$ and $B \geq 0$ for all $X$, we have that $\mathbb{P}(f(X) \geq M) \geq \mathbb{P}(f(X) > M) \geq \mathbb{P}(Bf(X) + h(X) > BM + h(X))$, where the probability is with respect to the randomness of $X$. Thus, by combining (4) and (5), we have that for any $h \in \mathcal{H}$, for any $\delta > 0$, with probability at least $1 - \delta$, the following holds for all $h \in \mathcal{H}$,

$$\mathbb{E}_r[\ell(h, r)] - \frac{1}{n} \sum_{i=1}^{n} \ell(h, r_i) \leq \frac{1}{n} \sum_{k=1}^{K} |\mathcal{I}_k| \left( \alpha_k(h) - \frac{1}{|\mathcal{I}_k|} \sum_{i \in \mathcal{I}_k} \ell(h, r_i) \right) + c\sqrt{\frac{\ln(2/\delta)}{n}}.$$

$\qquad\qquad\qquad\qquad\qquad\qquad\qquad\qquad\qquad\qquad\qquad\qquad\qquad\qquad\qquad\qquad\qquad\qquad\qquad\qquad\square$

In particular, the first term from the previous lemma will be bounded with the following lemma:

**Lemma 2.** *For any $f \in \{f_{\mathrm{SEM}(\tau)}^S, f_{\mathrm{base}}^S\}$,*

$$\frac{1}{n} \sum_{k=1}^{K} |\mathcal{I}_k| \left( \alpha_k(f) - \frac{1}{|\mathcal{I}_k|} \sum_{i \in \mathcal{I}_k} \ell(f, r_i) \right) \leq R\sqrt{L\varphi(f)}.$$

*Proof.* By using the triangle inequality,

$$\frac{1}{n} \sum_{k=1}^{K} |\mathcal{I}_k| \left( \mathbb{E}_r[\ell(f, r) | r \in \mathcal{C}_k] - \frac{1}{|\mathcal{I}_k|} \sum_{i \in \mathcal{I}_k} \ell(f, r_i) \right)$$

$$\leq \frac{1}{n} \sum_{k=1}^{K} |\mathcal{I}_k| \left| \mathbb{E}_r[\ell(f, r) | r \in \mathcal{C}_k] - \frac{1}{|\mathcal{I}_k|} \sum_{i \in \mathcal{I}_k} \ell(f, r_i) \right|.$$

Furthermore, by using the triangle inequality,

$$\left| \mathbb{E}_r[\ell(f, r) | r \in \mathcal{C}_k] - \frac{1}{|\mathcal{I}_k|} \sum_{i \in \mathcal{I}_k} \ell(f, r_i) \right| = \left| \frac{1}{|\mathcal{I}_k|} \sum_{i \in \mathcal{I}_k} \mathbb{E}_r[\ell(f, r) | r \in \mathcal{C}_k] - \frac{1}{|\mathcal{I}_k|} \sum_{i \in \mathcal{I}_k} \ell(f, r_i) \right|$$

$$\leq \frac{1}{|\mathcal{I}_k|} \sum_{i \in \mathcal{I}_k} \left| \mathbb{E}_r[\ell(f, r) | r \in \mathcal{C}_k] - \ell(f, r_i) \right|$$

$$\leq \sup_{r, r' \in \mathcal{C}_k} \left| \ell(f, r) - \ell(f, r') \right|.$$

If $f = f_{\mathrm{SEM}(\tau)}^S = g_{\mathrm{SEM}(\tau)}^S \circ \sigma_\tau$, since $g_{\mathrm{SEM}(\tau)}^S \in \mathcal{G}_S$, by using the Lipschitz continuity, boundedness, and non-negativity,

$$\sup_{r, r' \in \mathcal{C}_k} \left| \ell(f, r) - \ell(f, r') \right| = \sup_{y \in \mathcal{Y}} \sup_{z, z' \in \mathcal{Z}_k} |(l_y \circ g_{\mathrm{SEM}(\tau)}^S)(\sigma_\tau(z)) - (l_y \circ g_{\mathrm{SEM}(\tau)}^S)(\sigma_\tau(z'))|$$

$$\leq R \sup_{z, z' \in \mathcal{Z}_k} \|\sigma_\tau(z) - \sigma_\tau(z')\|_F$$

$$= R \sup_{z, z' \in \mathcal{Z}_k} \sqrt{\sum_{t=1}^{L} \sum_{j=1}^{V} (\sigma_\tau(z_{t,j}) - \sigma_\tau(z'_{t,j}))_2^2}$$

$$\leq R \sqrt{\sum_{t=1}^{L} \sup_{i \in [V]} \sup_{q, q' \in Q_i} \|\sigma_\tau(q) - \sigma_\tau(q')\|_2^2}$$

$$= R \sqrt{L\varphi(f_{\mathrm{SEM}(\tau)}^S)}$$

Similarly, if $f = f_{\text{base}}^S = g_{\text{base}}^S$, since $g_{\text{base}}^S \in \mathcal{G}_S$, by using the Lipschitz continuity, boundedness, and non-negativity,

$$
\sup_{r,r' \in \mathcal{C}_k} \left| \ell(f,r) - \ell(f,r') \right| = \sup_{y \in \mathcal{Y}} \sup_{z,z' \in \mathcal{Z}_k} \left| (l_y \circ g_{\text{base}}^S)(z) - (l_y \circ g_{\text{base}}^S)(z') \right|
$$
$$
\leq R \sup_{z,z' \in \mathcal{Z}_k} \|z - z'\|_F
$$
$$
\leq R \sqrt{L \varphi(f_{\text{base}}^S)}.
$$

Therefore, for any $f \in \{f_{\text{SEM}(\tau)}^S, f_{\text{base}}^S\}$,

$$
\frac{1}{n} \sum_{k=1}^{K} |\mathcal{I}_k| \left( \alpha_k(f) - \frac{1}{|\mathcal{I}_k|} \sum_{i \in \mathcal{I}_k} \ell(f,r_i) \right) \leq \frac{1}{n} \sum_{k=1}^{K} |\mathcal{I}_k| R \sqrt{L \varphi(f)} = R \sqrt{L \varphi(f)}.
$$

$\square$

Combining Lemma 1 and Lemma 2, we obtain the following upper bound on the gap:

**Lemma 3.** *For any $\delta > 0$, with probability at least $1 - \delta$, the following holds for any $f \in \{f_{\text{SEM}(\tau)}^S, f_{\text{base}}^S\}$:*

$$
\mathbb{E}_r[\ell(f,r)] - \frac{1}{n} \sum_{i=1}^{n} \ell(f,r_i) \leq R \sqrt{L \varphi(f)} + c \sqrt{\frac{\ln(2/\delta)}{n}}.
$$

*Proof.* This follows directly from combining Lemma 1 and Lemma 2. $\square$

We now provide an upper bound on $\varphi(f_{\text{SEM}(\tau)}^S)$ in the following lemma:

**Lemma 4.** *For any $\tau > 0$,*

$$
\varphi(f_{\text{SEM}(\tau)}^S) \leq \left| \frac{1}{1 + (V-1)e^{-2/\tau}} - \frac{1}{1 + (V-1)e^{-\Delta/\tau}} \right|^2
$$
$$
+ (V-1) \left| \frac{1}{1 + e^{\Delta/\tau}(1 + (V-2)e^{-2/\tau})} - \frac{1}{1 + e^{2/\tau}(1 + (V-2)e^{-\Delta/\tau})} \right|^2.
$$

*Proof.* Recall the definition:

$$
\varphi(f_{\text{SEM}(\tau)}^S) = \sup_{i \in [V]} \sup_{q,q' \in Q_i} \|\sigma_\tau(q) - \sigma_\tau(q')\|_2^2.
$$

where

$$
\sigma_\tau(q)_j = \frac{e^{q_j/\tau}}{\sum_{t=1}^{V} e^{q_t/\tau}},
$$

for $j = 1, \ldots, V$. By the symmetry and independence over $i \in [V]$ inside of the first supremum, we have

$$
\varphi(f_{\text{SEM}(\tau)}^S) = \sup_{q,q' \in Q_1} \|\sigma_\tau(q) - \sigma_\tau(q')\|_2^2.
$$

For any $q, q' \in Q_1$ and $i \in \{2, \ldots, V\}$ (with $q = (q_1, \ldots, q_V)$ and $q' = (q_1', \ldots, q_V')$), there exists $\delta_i, \delta_i' > 0$ such that

$$
q_i = q_1 - \delta_i
$$

and

$$
q_i' = q_1' - \delta_i'.
$$

Here, since $z_{ik} - \Delta \geq z_{ij}$ from the assumption, we have that for all $i \in \{2, \ldots, V\}$,

$$
\delta_i, \delta_i' \geq \Delta > 0.
$$

Thus, we can rewrite

$$\sum_{t=1}^{V} e^{q_t/\tau} = e^{q_1/\tau} + \sum_{i=2}^{V} e^{(q_1-\delta_i)/\tau}$$

$$= e^{q_1/\tau} + e^{q_1/\tau} \sum_{i=2}^{V} e^{-\delta_i/\tau}$$

$$= e^{q_1/\tau} \left( 1 + \sum_{i=2}^{V} e^{-\delta_i/\tau} \right)$$

Similarly,

$$\sum_{t=1}^{V} e^{q'_t/\tau} = e^{q'_1/\tau} \left( 1 + \sum_{i=2}^{V} e^{-\delta'_i/\tau} \right).$$

Using these,

$$\sigma_\tau(q)_1 = \frac{e^{q_1/\tau}}{\sum_{t=1}^{V} e^{q_t/\tau}} = \frac{e^{q_1/\tau}}{e^{q_1/\tau} \left( 1 + \sum_{i=2}^{V} e^{-\delta_i/\tau} \right)} = \frac{1}{1 + \sum_{i=2}^{V} e^{-\delta_i/\tau}}$$

and for all $j \in \{2, \ldots, V\}$,

$$\sigma_\tau(q)_j = \frac{e^{q_j/\tau}}{\sum_{t=1}^{V} e^{q_t/\tau}}$$

$$= \frac{e^{(q_1-\delta_j)/\tau}}{e^{q_1/\tau} \left( 1 + \sum_{i=2}^{V} e^{-\delta_i/\tau} \right)}$$

$$= \frac{e^{-\delta_j/\tau}}{1 + \sum_{i=2}^{V} e^{-\delta_i/\tau}}$$

$$= \frac{1}{1 + e^{\delta_j/\tau} + \sum_{i\in I_j}^{V} e^{(\delta_j-\delta_i)/\tau}}$$

where $I_j := \{2, \ldots, V\} \setminus \{j\}$. Similarly,

$$\sigma_\tau(q')_1 = \frac{1}{1 + \sum_{i=2}^{V} e^{-\delta'_i/\tau}},$$

and for all $j \in \{2, \ldots, V\}$,

$$\sigma_\tau(q')_j = \frac{1}{1 + e^{\delta'_j/\tau} + \sum_{i\in I_j}^{V} e^{(\delta'_j-\delta'_i)/\tau}}.$$

Using these, for any $q, q' \in Q_1$,

$$|\sigma_\tau(q)_1 - \sigma_\tau(q')_1| = \left| \frac{1}{1 + \sum_{i=2}^{V} e^{-\delta_i/\tau}} - \frac{1}{1 + \sum_{i=2}^{V} e^{-\delta'_i/\tau}} \right|$$

$$\leq \left| \frac{1}{1 + \sum_{i=2}^{V} e^{-2/\tau}} - \frac{1}{1 + \sum_{i=2}^{V} e^{-\Delta/\tau}} \right|$$

$$= \left| \frac{1}{1 + (V-1)e^{-2/\tau}} - \frac{1}{1 + (V-1)e^{-\Delta/\tau}} \right|,$$

and for all $j \in \{2, \ldots, V\}$,

$$|\sigma_\tau(q)_j - \sigma_\tau(q')_j| = \left| \frac{1}{1 + e^{\delta_j/\tau} + \sum_{i \in I_j}^V e^{(\delta_j - \delta_i)/\tau}} - \frac{1}{1 + e^{\delta'_j/\tau} + \sum_{i \in I_j}^V e^{(\delta'_j - \delta'_i)/\tau}} \right|$$

$$\leq \left| \frac{1}{1 + e^{\Delta/\tau} + \sum_{i \in I_j}^V e^{(\Delta - 2)/\tau}} - \frac{1}{1 + e^{2/\tau} + \sum_{i \in I_j}^V e^{(2 - \Delta)/\tau}} \right|$$

$$= \left| \frac{1}{1 + e^{\Delta/\tau} + (V - 2)e^{(\Delta - 2)/\tau}} - \frac{1}{1 + e^{2/\tau} + (V - 2)e^{(2 - \Delta)/\tau}} \right|$$

$$= \left| \frac{1}{1 + e^{\Delta/\tau}(1 + (V - 2)e^{-2/\tau})} - \frac{1}{1 + e^{2/\tau}(1 + (V - 2)e^{-\Delta/\tau})} \right|.$$

By combining these,

$$\sup_{q,q' \in Q_1} \|\sigma_\tau(q) - \sigma_\tau(q')\|_2^2$$

$$= \sup_{q,q' \in Q_1} \sum_{j=1}^V |\sigma_\tau(q)_j - \sigma_\tau(q')_j|^2$$

$$\leq \left| \frac{1}{1 + (V - 1)e^{-2/\tau}} - \frac{1}{1 + (V - 1)e^{-\Delta/\tau}} \right|^2$$

$$+ (V - 1) \left| \frac{1}{1 + e^{\Delta/\tau}(1 + (V - 2)e^{-2/\tau})} - \frac{1}{1 + e^{2/\tau}(1 + (V - 2)e^{-\Delta/\tau})} \right|^2.$$

$\square$

Using the previous lemma, we will conclude the asymptotic behavior of $\varphi(f^S_{\text{SEM}(\tau)})$ in the following lemma:

**Lemma 5.** *It holds that*

$$\varphi(f^S_{\text{SEM}(\tau)}) \to 0 \text{ as } \tau \to 0.$$

*Proof.* Using Lemma 4,

$$\lim_{\tau \to 0} \varphi(f^S_{\text{SEM}(\tau)}) \leq \lim_{\tau \to 0} \left| \frac{1}{1 + (V - 1)e^{-2/\tau}} - \frac{1}{1 + (V - 1)e^{-\Delta/\tau}} \right|^2$$

$$+ n(V - 1) \lim_{\tau \to 0} \left| \frac{1}{1 + e^{\Delta/\tau}(1 + (V - 2)e^{-2/\tau})} - \frac{1}{1 + e^{2/\tau}(1 + (V - 2)e^{-\Delta/\tau})} \right|^2.$$

Moreover,

$$\lim_{\tau \to 0} \left| \frac{1}{1 + (V - 1)e^{-2/\tau}} - \frac{1}{1 + (V - 1)e^{-\Delta/\tau}} \right|^2 = \left| \frac{1}{1} - \frac{1}{1} \right|^2 = 0,$$

and

$$\lim_{\tau \to 0} \left| \frac{1}{1 + e^{\Delta/\tau}(1 + (V - 2)e^{-2/\tau})} - \frac{1}{1 + e^{2/\tau}(1 + (V - 2)e^{-\Delta/\tau})} \right|^2 = |0 - 0|^2 = 0.$$

Therefore,

$$\lim_{\tau \to 0} \varphi(f^S_{\text{SEM}(\tau)}) \leq 0.$$

Since $\varphi(f^S_{\text{SEM}(\tau)}) \geq 0$, this implies the statement of this lemma. $\square$

As we have analyzed $\varphi(f^S_{\mathrm{SEM}(\tau)})$ in the previous two lemmas, we are now ready to compare $\varphi(f^S_{\mathrm{SEM}(\tau)})$ and $\varphi(f^S_{\mathrm{base}})$, which is done in the following lemma:

**Lemma 6.** *For any $\tau > 0$,*

$$\varphi(f^S_{\mathrm{SEM}(\tau)}) - \varphi(f^S_{\mathrm{base}}) \leq \frac{3}{4}(1 - V) < 0.$$

*Proof.* From Lemma 4, for any $\tau > 0$,

$$\varphi(f^S_{\mathrm{SEM}(\tau)}) \leq \left| \frac{1}{1 + (V-1)e^{-2/\tau}} - \frac{1}{1 + (V-1)e^{-\Delta/\tau}} \right|^2$$

$$+ n(V-1) \left| \frac{1}{1 + e^{\Delta/\tau}(1 + (V-2)e^{-2/\tau})} - \frac{1}{1 + e^{2/\tau}(1 + (V-2)e^{-\Delta/\tau})} \right|^2$$

$$\leq \left| \frac{1}{1 + (V-1)e^{-2/\tau}} - \frac{1}{1 + (V-1)} \right|^2$$

$$+ (V-1) \left| \frac{1}{1 + (1 + (V-2)e^{-2/\tau})} - \frac{1}{1 + e^{2/\tau}(1 + (V-2))} \right|^2$$

$$= \left| \frac{1}{1 + (V-1)e^{-2/\tau}} - \frac{1}{V} \right|^2 + (V-1) \left| \frac{1}{2 + (V-2)e^{-2/\tau}} - \frac{1}{1 + e^{2/\tau}(V-1)} \right|^2$$

$$\leq \left| \frac{1}{1} - \frac{1}{V} \right|^2 + (V-1) \left| \frac{1}{2} - 0 \right|^2$$

$$= \left( \frac{1}{1} - \frac{1}{V} \right)^2 + (V-1)\frac{1}{4}.$$

Recall the definition of

$$\varphi(f^S_{\mathrm{base}}) = \sup_{i \in [V]} \sup_{q,q' \in Q_i} \|q - q'\|_2^2.$$

By choosing an element in the set over which the supremum is taken, for any $\delta \geq \Delta > 0$,

$$\varphi(f^S_{\mathrm{base}}) \geq \sup_{q,q' \in Q_1} \|q - q'\|_2^2 \geq \|\hat{q} - \hat{q}'\|_2^2 = \sum_{j=1}^{V} (\hat{q}_j - \hat{q}'_j)_2^2 = (2 - \delta)^2 V,$$

where $\hat{q}_1 = 1$, $\hat{q}_j = 1 - \delta$ for $j \in \{2, \ldots, V\}$, $\hat{q}'_1 = \delta - 1$, and $\hat{q}'_j = -1$ for $j \in \{2, \ldots, V\}$.

By combining those, for for any $\tau > 0$ and $\delta \geq \Delta > 0$,

$$\varphi(f^S_{\mathrm{SEM}(\tau)}) - \varphi(f^S_{\mathrm{base}}) \leq \left( \frac{1}{1} - \frac{1}{V} \right)^2 + (V-1)\frac{1}{4} - (2-\delta)^2 V$$

$$\leq 1 + \frac{1}{4}V - \frac{1}{4} - (2-\delta)^2 V$$

$$= \frac{3}{4} + \frac{1}{4}V - (2-\delta)^2 V$$

$$= \frac{3}{4} - V \left( (2-\delta)^2 - \frac{1}{4} \right)$$

$$\leq \frac{3}{4} - V \left( 1 - \frac{1}{4} \right)$$

$$= \frac{3}{4}(1 - V)$$

$\square$

We combine the lemmas above to prove Theorem 1, which is restated below with its proof:

**Theorem 1.** *Let $V \geq 2$. For any $1 \geq \delta > 0$, with probability at least $1 - \delta$, the following holds for any $f_S \in \{f_{\mathrm{SEM}(\tau)}^S, f_{\mathrm{base}}^S\}$:*

$$\mathbb{E}_{z,y}[l(f_S(z), y)] \leq \frac{1}{n} \sum_{i=1}^{n} l(f_S(z^{(i)}), y^{(i)}) + R\sqrt{L\,\varphi_{f_S}(V, \tau)} + c\sqrt{\frac{\ln(2/\delta)}{n}},$$

*where $c > 0$ is a constant in $(n, f, \mathcal{H}, \delta, \tau, S)$. Moreover,*

$$\varphi_{f_{\mathrm{SEM}(\tau)}^S} \to 0 \quad as \ \tau \to 0 \quad and \quad \varphi_{f_{\mathrm{SEM}(\tau)}^S} - \varphi_{f_{\mathrm{base}}^S} \leq \frac{3}{4}(1 - V) < 0 \quad \forall \tau > 0.$$

*Proof.* The first statement directly follows from Lemma 3. The second statement is proven by Lemma 5 and Lemma 6. □

## B EXPERIMENT DETAILS FOR IMAGENET

### B.1 IMAGE AUGMENTATION

The augmentation applied in order during training are:

- Random Resize crop to a $224 \times 224$ image. A random patch of the image is selected and resized to a $224 \times 224$ image.
- Random color jitter. Modifying the brightness, the contrast, the saturation and the hue.
- Random gray scale. Randomly applying a gray scale filter to the image
- Random Gaussian blur. Randomly applying a Gaussian bluer filter.
- Random solarization. Randomly applying a solarization filter.

The parameters of the augmentations are presented in Table 16. At validation and test time, we resize the images to $256 \times 256$ and then center crop a patch of $224 \times 224$.

For both training and evaluation, we re-normalize the image using the statistic of the training set.g

### B.2 LINEAR EVALUATION

We follow the evaluation protocol from (Chen et al., 2020b). The linear evaluation is done by training a linear classifier on the frozen representation of the ImageNet training samples. We train a linear classifier with a cross-entropy objective for 100 epochs using SGD with nesterov, a momentum of $0.9$ and a batch size of 256. We perform learning rate scheduling at epoch 60 and epoch 80 where we divide the learning rate by a factor of 10. During training, we apply random resized crop to $224 \times 224$ pixels and random horizontal flip. We sweep over a set of 4 learning rates: $\{0.5, 0.1, 0.05, 0.01\}$, 3 $l1$ weight decays: $\{0, 1e-6, 1e-5\}$ and 3 $\tau_d$ for SEM: $\{0.01, 0.1, 1\}$, using a validation set of 10 images per class and re-traing using the full training set. We report the results on the test set.

### B.3 ROBUSTNESS EXPERIMENTS

We follow the evaluation procedure from (Lee et al., 2021). We treated the robustness datasets as additional "test sets" in that we simply evaluated them using the evaluation procedure described above. The images were resized to a $256 \times 256$ before being center cropped to a $224 \times 224$ image. The evaluation procedure was performed using the public robustness benchmark evaluation code of (Djolonga et al., 2020)[§].

### B.4 TRANSFER LEARNING LINEAR PROBE

We follow the linear evaluation protocol of (Kolesnikov et al., 2019; Chen et al., 2020b) We train a linear classifier using a regularized multinomial logistic regression from the scikit-learn package (Pedregosa et al., 2011). The representation is frozen, so that we do not train the encoder backbone nor

---

[§]https://github.com/google-research/robustness_metrics

the batch-normalization statistics. We do not perform any augmentations and the images are resized to 224 pixels using bicubic resampling and the normalized using the statistics on ImageNet's training set. We tune the regularizer term from a range of 45 logarithmically-spaced values between $10^{-6}$ and $10^5$ using a small validation set and re-train using the full training set. For SEM, we set $\tau_d = 0$ for all experiments.

### B.5 TRANSFER LEARNING FINE-TUNING

We follow the same fine-tuning protocol of (Chen et al., 2020b; Grill et al., 2020). We initialize the encoder with the pre-trained model and a classifier head with random initialization. We train for 20,000 steps with a batch size of 256 using SGD with a Nesterov momentum of 0.9. We set the momentum parameter for the batch normalization to be $\max(1 - 10/s, 0.9)$ where $s$ is the number of steps per epoch. During pre-training, we use random resize to $224 \times 224$ pixels and random horizontal flipping. At test time, we resize the images along the shortest size to 256 pixels using cubic resampling following by a center resize to $224 \times 224$ pixels. Due to computational constraint, we only tune the learning rate using a search of 7 values spaces on logarithmic scales between 0.0001 and 0.1. For SEM, we set $\tau_d = 1$. for all experiments After choosing the best learning rate of a validation set, we re-run the models using the full training set and evaluate it on the test set, which we use to report the numbers.

### B.6 SEMI-SUPERVISED LEARNING

We follow the semi-supervised learning protocol of (Chen et al., 2020b; Grill et al., 2020). We initialize the network using the pre-trained representation and initialize a classification head using random initialization. We fine-tune the encoder while training the classification head using a small subset of ImageNet. We choose the same subset used in prior works which is defined in the TensorFlow-Dataset software. During training, we random resize the images to $224 \times 224$ pixels along the shorter size using bicubic resampling followed by a center crop and random horizontal flipping. At test time, we resize the image to $224 \times 224$. We optimize the cross entropy loss with nestorov and a momentum of 0.9 using batch sizes of 224. We train models for $\{30, 50\}$ and take the best performing on the validation set. The learning rate used is chosen among a set of 5 learning rates: $\{0.01, 0.02, 0.05, 0.1, 0.005\}$. For SEM, we also search $\tau_d \in \{0.01, 0.1, 1\}$. We perform the search on the best performing one on the validation set and the number are returned are obtained using the test set after re-training using the full training set.

## C HYPERPARAMETERS

The implementation of the SSL methods used in this work are taken from Solo-Learn (da Costa et al., 2021) to which we added the SEM module. The pre-training hyper-parameters of every SSL methods trained on CIFAR-100 with ResNet-18 used in this work are the default provided in the companion repository of Solo-Learn. The hyper-parameters are also provided in the launch scripts accompanying this work. Due to the large number of SSL methods probed in this work and the amount of space it would require to exhaustively detail all of the hyper-parameters, we refer the reader to the code.

For the CIFAR-100 results obtained with BYOL and a ResNet-50, we have slightly modified the default parameters. Otherwise, the baseline BYOL model would not obtain competitive results. The hyper-parameters were tuned using the BYOL baseline and the SEM module was not considered in the selection of the SSL hyper-parameters. The BYOL hyper-parameters are presented in the launch script accompanying this work and presented below for completeness.

For the ImageNet experiments, we took the hyper-parameters proposed in the launch scripts of Solo-Learn to which we only modified the amount of epochs (100 epochs to 200 epochs.)

Here, we present all of the SEM hyper-parameters used in every experiments. These hyper-parameters can also be found in the launch scripts accompanying this work.

We present the hype-parameters used to train for BYOL+SEM and MoCo+SEM on CIFAR100. Unless mentioned otherwise, these are the parameters used.

Table 6: BYOL with ResNet-50 for CIFAR-100.

| | |
|---|---|
| precision | 16 |
| Learning rate | 0.5 |
| Weight-decay | 1e-4 |
| Optimizer | sgd + lars |
| LR scheduler | warmup + cosine |
| eta lars | 0.001 |
| exclude bias n norm (lars) | True |
| batch size | 256 |
| base ema momentum | 0.99 |
| final ema momentum | 1.0 |
| proj output dim | 256 |
| proj hidden dim | 4096 |
| pred hidden dim | 4096 |
| *augmentations*: | |
| solarization_prob | view 1: 0 view 2: 0.2 |
| crop size | 32 |
| hue | 0.1 |
| saturation | 0.2 |
| contrast | 0.4 |
| brightness | 0.4 |

Table 7: SEM SimCLR RN-18 for CIFAR-100

| L | V | $\tau_p$ | $\tau_p'$ |
|---|---|---|---|
| 5000 | 13 | 0.17 | 0.78 |

Table 8: SEM MoCo RN-18 for CIFAR-100

| L | V | $\tau_p$ | $\tau_p'$ |
|---|---|---|---|
| 5000 | 13 | 0.04 | 0.01 |

Table 9: SEM BYOL RN-18 for CIFAR-100

| L | V | $\tau_p$ | |
|---|---|---|---|
| 5000 | 13 | 1.0 | 1.0 |

Table 10: SEM SwAV RN-18 for CIFAR-100

| L | V | $\tau_p$ | $\tau_p'$ |
|---|---|---|---|
| 5000 | 13 | 0.85 | 1.5 |

Table 11: SEM DINO RN-18 for CIFAR-100

| L | V | $\tau_p$ | $\tau_p'$ |
|---|---|---|---|
| 5000 | 13 | 1.0 | 1.0 |

Table 12: SEM Barlow RN-18 for CIFAR-100

| L | V | $\tau_p$ | $\tau_p'$ |
|---|---|---|---|
| 5000 | 13 | 1.0 | 0.99 |

Table 13: SEM VicREG RN-18 for CIFAR-100

| L | V | $\tau_p$ | $\tau_p'$ |
|---|---|---|---|
| 5000 | 13 | 1.0 | 1.0 |

Table 14: SEM BYOL RN-50 for CIFAR-100

| L | V | $\tau_p$ | $\tau_p'$ |
|---|---|---|---|
| 5000 | 13 | 1 | 1 |

Table 15: SEM BYOL all ResNets for ImageNet

| L | V | $\tau_p$ | $\tau_p'$ |
|---|---|---|---|
| 5000 | 21 | 0.16 | 0.04 |

## C.1 COMPUTATIONAL RESOURCES

For all our CIFAR-100 training, we used 1 RTX-8000 per experiment. For our ImageNet experiments, we used parallel training with 2 40GB A100 for the training with ResNet50 and ResNet50-x2 and 4 40GB A100 for the training with ResNet50-x4. With this setup, the training takes about a week for the ResNet50 experiments and about 10 days for the ResNet50-x2 and ResNet50-x4 experiments.

## D ADDITIONAL STUDIES OF SEM

In Section 4.2, we discussed the effect of scaling $L$ and $V$ as well as changing the Softmax temperature during pre-training of the online network and changing the Softmax temperature for the downstream task. Here, we propose additional studies of SEM to provide a better mastery of the method. We provide a method for reducing the memory overhead of SEM and experiments demonstrating that despite this version still largely outperform the baseline. We additionally present the effect of modifying the embedder contributing to the insight on how to get the most out of SEM. Next, we have discussion with a study of the spectrum of the covariance matrix of the SEM representation and the BYOL representation, showing insight how SEM can particularly improve the training signal

Table 16: BYOL with all ResNet-50 architectures for ImageNet.

| | |
|---|---|
| precision | 16 |
| Learning rate | 0.4 |
| Weight-decay | 1e-6 |
| Optimizer | sgd + lars |
| LR scheduler | warmup + cosine |
| eta lars | 0.001 |
| exclude bias n norm (lars) | True |
| batch size | 256 |
| base ema momentum | 0.99 |
| final ema momentum | 1.0 |
| proj output dim | 256 |
| proj hidden dim | 4096 |
| pred hidden dim | 4096 |
| *augmentations*: | |
| solarization_prob | view 1: 0 view 2: 0.2 |
| gaussian_prob | view 1: 1.0 view 2: 0.1 |
| crop size | 224 |
| hue | 0.1 |
| saturation | 0.2 |
| contrast | 0.4 |
| brightness | 0.4 |

Table 17: # of parameters, # of activations, allocated memory, computation efficiency (FLOPs/sample) and CIFAR-100 accuracy of BYOL, BYOL with SEM and its memory-efficient variant with $8$ blocks (denoted BYOL + SEM/8).

| | # params | # activations | vRAM (GiB) | FLOPs | Accuracy |
|---|---|---|---|---|---|
| *Resnet-18*: | | | | | |
| BYOL | $16.5M$ | $0.731M$ | 4.0 | $7.20e8$ | 70.7 |
| BYOL+SEM | $313.7M$ | $0.797M$ | 13.1 | $1.01e9$ | 73.9 |
| BYOL+SEM/8 | $51.9M$ | $0.796M$ | 5.3 | $7.46e8$ | 73.3 |
| *Resnet-50*: | | | | | |
| BYOL | $35M$ | $4.05M$ | 11.1 | $1.65e9$ | 74.3 |
| BYOL+SEM | $425.6M$ | $4.12M$ | 21.9 | $2.04e9$ | 77.4 |
| BYOL+SEM/8 | $76.7M$ | $4.12M$ | 11.8 | $1.69e9$ | 76.6 |

during pre-training. We provide a scaling analysis of BYOL and BYOL + SEM on CIFAR-100. We end with an experiment showing that pre-training with SEM is necessary to get the best performance.

## D.1 An efficient variant of SEM

A large over-complete representation may induce a significant memory footprint due to the additional parameters of the fully connected linear layer used to map to and from the representation. For SEM we require two such mappings as depicted in Figure 2c for BYOL. To reduce the amount of parameters, we propose to sparsify the weight matrix of the fully connected linear layer. We propose to do so by taking the block diagonal of the parameters of the matrix multiplication and setting the parameters outside the block diagonal to 0. Formally, let $v \in \mathbb{R}^{b \times m}$, $w \in \mathbb{R}^{m \times o}$ and $y = v \cdot w$ be the fully connected matrix multiplication. Instead, we partition $v$ into $n$ blocks with $v^i \in \mathbb{R}^{b \times \frac{m}{n}}$ and define $n$ smaller $w^i \in \mathbb{R}^{\frac{m}{n} \times \frac{o}{n}}$, where $i \in [L]$ is the $i^{th}$ block. Then, we perform a batch matrix multiplication of $v^i$ and $w^i$ that we concatenate as follows: $y^i = v^i \cdot w^i$ and $\bar{y}^i = \text{Concat}([y^1, \dots, y^n])$. Thus, the amount of parameters of this matrix multiplication scales in $\mathcal{O}(\frac{m \cdot o}{n})$, allowing us to reduce the memory consumption by increasing $n$, the number of blocks.

We perform an experiment where we partition the embedder and the first linear layer of the projector into $8$ blocks. We present the results in Table 17 in which we compare the # of parameters, the

# of activations, the allocated vRAM by pytorch, the FLOPs/sample and the accuracy of BYOL, BYOL+SEM and BYOL+SEM/8 representing the model with 8 blocks obtained following the method described above. We observe that partitioning the matrix multiplications of SEM allows to vastly reduce the computation parameters while still yielding an important improvement over the baseline. This result demosntrate that SEM can be beneficial while inducing minimal computational overhead.

Attentive readers may notice that this performance is better compared to the ablation presented in Figure 3. The difference in performance is due to probing the embedder's output (i.e. $z_\theta$) in Figure 3 and probing the encoder's output (i.e. $e_\theta$) in Table 17. Using the each ablation's representation for probing to the other recovers the performance observed by each.

## D.2   ADDITIONAL ABLATION OF THE SEM PARAMETERS

**Ablating the embedder**   In the main text, we mentioned that we use batch normalization)) at the output of the embedder. The reason we use batch normalization is mostly due to the fact that we wanted to avoid tuning any hyper-parameters that were not related to SEM to emphasize its contribution. Using BatchNorm gave the best performance without tuning the hyper-parameters of the baseline models.

Here, we want to emphasize that SEM can be used without batch norm, but more hyper-parameters might need to be tuned for it to perform as well as the model with batch norm in the encoder. For example, we found that using no weight decay was important to get better performance when we did not have batch normalization as illustrated in Table table 18. We leave the full study of the interaction of SEM with the SSL related parameters for future work.

Table 18: Understanding the relationship between the use of BatchNorm in the embedder and the weight decay hyper-parameter.

| BatchNorm | weight decay | Accuracy |
|:---:|:---:|:---:|
|  | 0 | 67.2 |
|  | 1e-5 | 57.9 |
| ✓ | 0 | 68.3 |
| ✓ | 1e-5 | 73.9 |

Another decision is to use a linear layer as the embedder. Other alternative may include using the Identidy function (i.e. the output of the encoder is used for SEM). However, if we want to systematically use the same encoder as the SSL model, then we are constrained to a representation size that is the one of the ResNet encoder (i.e. 512 for a ResNet-18).

Finally, we showcase that using a more expressive embedder leads to exacerbated performance and recommend practitioner to limit the expressivity of their embedder.

Table 19: Comparing alternative embedders.

|  | Accuracy |
|:---:|:---:|
| Identity | 63.0 |
| Linear | 73.9 |
| 1 hidden layer MLP | 65.0 |

**A very very large embedding**   Using a ResNet-18 encoder and the method proposed in Section D.1, we further scale the embedding size of SEM to see where the performance saturates for classification. In Figure 7 we observe that the performance saturates for $L = 10000$ for the classification task. We conjecture that the optimal $L$ might be different for other tasks, but we leave that study for future work.

## D.3   ANALYZE OF THE SPECTRUM OF THE COVARIANCE MATRIX OF THE REPRESENTATION

To obtain a better insight on why the SEM representation leads to better downstream performance, we analyze the spectrum of the covariance matrix of the representation using the methodology presented

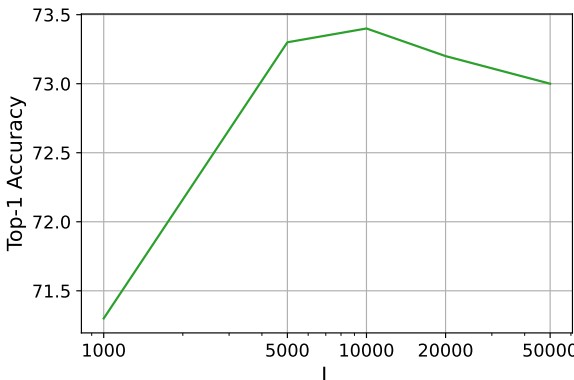

Figure 7: Study of very very large $L$ using a ResNet-18 backbone and $8$ SEM/8 blocks using the method described in Section D.1.

in Jing et al. (2022). That is, we collect the embedding vectors of the test set of CIFAR-100 using a pre-trained model using ResNet-50. For BYOL, we have an additional embedder without softmax normalization (as done in Figure 3). For BYOL and BYOL+SEM we use the embedder's output $(z_\theta)$ to perform the evaluation. To compute the covariance matrix $C \in \mathcal{R}^{L \cdot V \times L \cdot V}$ of the embedding layer $z$, we define $\bar{z} := \sum_{i=1}^{N} z_i/N$ the average representation over the N samples and compute the covariance as follows:

$$C := \frac{1}{N} \sum_{i=1}^{N} (z_i - \bar{z})(z_i - \bar{z})^\top. \tag{6}$$

To plot the spectrum of the covariance matrix, we take the singularalue decomposition of the matrix $(C = USV^\top)$ with S the diagonal of the singular values, which we plot in sorted order and logarithm scale in Figure 8.

This experiment demonstrates that the softmax normalization counters the dimensionality collapse that was discussed in Jing et al. (2022). Interestingly, the drop observed with SEM with $L \geq 500$ occurs at the index 2048 which is the dimensionality output of the ResNet-50 encoder.

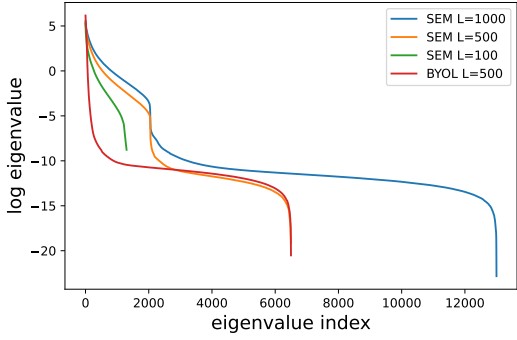

Figure 8: Spectrum of the covariance matrix of the represention for BYOL and BYOL + SEM obtained with a ResNet-50 encoder.

### D.4 SCALING THE RESNET ENCODER FOR CIFAR-100

We perform a scaling experiment on CIFAR-100 where we compare the scaling behaviour of BYOL and BYOL + SEM. We evaluate the computational cost of the methods and the resulting downstream accuracy for a range of four resnets: ResNet-18, ResNet-50, ResNet-50 x2 and ResNet-50 x4. In Figure 9, we observe that SEM has a better scaling behaviour than the baseline, especially as we increase the width of the ResNet-50. For BYOL, we observe that the performance decays for

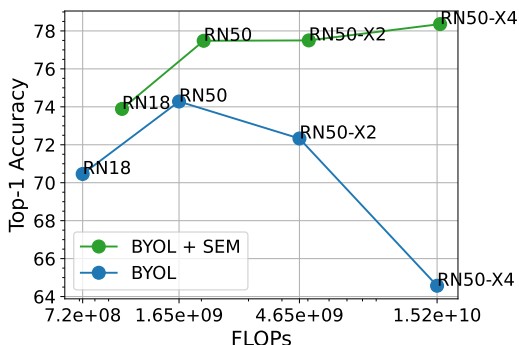

Figure 9: Scaling the ResNet encoder for CIFAR-100.

Table 20: Downstream accuracy of training a classifier with SEM normalization of the representation while using unnormalized representation during pretraining. Experiments performed with a ResNet-50 encoder.

| Pre-train model | Probe location | SEM($\tau = 0.1$) | Accuracy |
|---|---|---|---|
| BYOL + Embed | Embedder | No | 69.8 |
| BYOL + Embed | Embedder | Yes | 72.3 |
| BYOL + SEM | Embedder | Yes | 77.3 |

ResNet-50 with width x2 and x4. This is not unprecedented, as prior works as demonstrated other methods where scaling up the capacity of a model led to decrease in performance. When comparing the discrepancy with Figure 1, we attribute that to the fact that CIFAR-100 is a small dataset. In fact, we observe that the training accuracy stays constant to about 79% for all the ResNet-50 scales demonstrating overfitting for the baseline BYOL. Nevertheless, SEM prevents the decrease in performance and even lead to further improved performance as we increase the scale of the ResNet-50.

## D.5 THE ROLE OF PRE-TRAINING WITH SEM

We probe the downstream accuracy obtained of a model pre-trained without SEM and add SEM normalization only for the downstream classification. For this experiment, we take a pre-trained model with embedder (i.e. BYOL + embed) with $L = 5000$ and $V = 13$ and add the softmax normalization only for the downstream classification. We do not use SEM during pre-training. We observe that using SEM for downstream classification leads to an improvement even when the model is not pre-trained with SEM, demonstrating the utility of SEM downstream classification. However, we note that the performance of the model pre-trained without SEM is much weaker and thus demonstrates the imprtance of also pre-training using SEM.

## E  CIFAR-10 RESULTS

We confirm that our method also yield improvement on simpler datasets such as CIFAR-10. Here, we compare BYOL and BYOL + SEM on a ResNet-50 and observe and improvement of 1.6%.

Table 21: Downstream accuracy of training a classifier with SEM normalization of the representation while using unnormalized representation during pretraining. Experiments performed with a ResNet-50 encoder.

| Pre-train model | TOP-1 Accuracy |
|---|---|
| BYOL | 94.2 |
| BYOL + SEM | 95.8 |

## F  CIFAR100 SUPERCLASS

The 100 classes of CIFAR-100 (Krizhevsky, 2009) are grouped into 20 superclasses. The list of superclass for each class in Table 22

Table 22: Set of classes for each superclass on CIFAR-100.

| Superclass | Classes |
|---|---|
| aquatic mammals | beaver, dolphin, otter, seal, whale |
| fish | aquarium fish, flatfish, ray, shark, trout |
| flowers | orchids, poppies, roses, sunflowers, tulips |
| food containers | bottles, bowls, cans, cups, plates |
| fruit and vegetables | apples, mushrooms, oranges, pears, sweet peppers |
| household electrical devices | clock, computer keyboard, lamp, telephone, television |
| household furniture | bed, chair, couch, table, wardrobe |
| insects | bee, beetle, butterfly, caterpillar, cockroach |
| large carnivores | bear, leopard, lion, tiger, wolf |
| large man-made outdoor things | bridge, castle, house, road, skyscraper |
| large natural outdoor scenes | cloud, forest, mountain, plain, sea |
| large omnivores and herbivores | camel, cattle, chimpanzee, elephant, kangaroo |
| medium-sized mammals | fox, porcupine, possum, raccoon, skunk |
| non-insect invertebrates | crab, lobster, snail, spider, worm |
| people | baby, boy, girl, man, woman |
| reptiles | crocodile, dinosaur, lizard, snake, turtle |
| small mammals | hamster, mouse, rabbit, shrew, squirrel |
| trees | maple, oak, palm, pine, willow |
| vehicles 1 | bicycle, bus, motorcycle, pickup truck, train |
| vehicles 2 | lawn-mower, rocket, streetcar, tank, tractor |

# G  ADDITIONAL CIFAR-100 COHERENCE GRAPHS

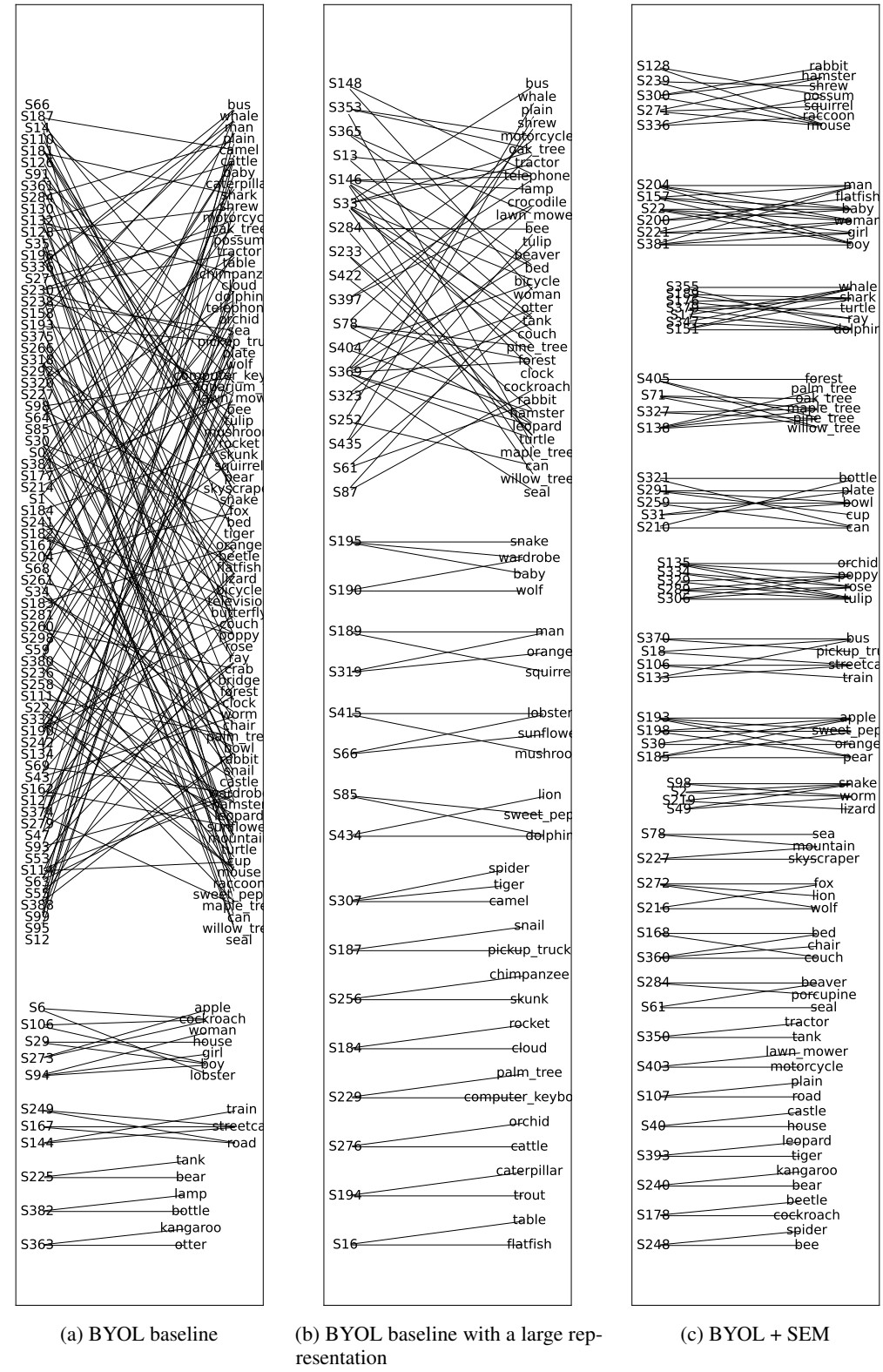

Figure 10: Comparison of the full semantic coherence graph $\mathcal{W}_5$ between BYOL and BYOL + SEM.

(a) BYOL baseline     (b) BYOL baseline with a large representation     (c) BYOL + SEM

