# OpenReview forum: "Simplicial Embeddings in Self-Supervised Learning and Downstream Classification"
_ICLR.cc/2023/Conference — ICLR 2023 notable top 25%_

### Official Review · Reviewer_Jv3d · 2022-10-21

**Confidence:** 3
**Correctness:** 4
**Technical Novelty And Significance:** 4
**Empirical Novelty And Significance:** 4
**Recommendation:** 8

**Clarity, Quality, Novelty And Reproducibility:**

Quality

This paper is really solid with regards to explaining the method and examining all the details of how to use it.  The experiments show benefits in many supervised learning tasks, but may need to use stronger baselines.  The theory is interesting but may not be practical.


Clarity

The paper is clearly written.


Originality

SEM is conceptually similar to previous approaches, but somehow performs much better than them.

**Strength And Weaknesses:**

Strengths

+ Simple idea with interesting connections to sparse coding (in neuroscience)
+ Goes beyond looking at performance and also shows improved interpretability.
+ Includes a theoretical analysis.
+ Provides useful guidance on setting hyperparameters.
+ Evaluations show benefits in several problems: classification (without and without corruption) and transfer learning.


Weaknesses

- Why are some of the competing methods doing so poorly?  Is the poor performance of BYOB+Gumbel on CIFAR-100 really due to SEM being a better method, or perhaps the Gumbel approach was not implemented as well as possible? [after rebuttal: addressed]
- It would be nice to how changing the architecture affects results. [after rebuttal: addressed]
- A minor weakness, but the use of softmax is so commonplace that it doesn't even seem to warrant a mention in related work.
- The practical utility of the theoretical result is unclear.
- As reviewer E9n7 and 4CQ8 noted, the baselines could be improved.  BYOL should be trained until convergence, and increasing the number of nonlinear layers or increasing the size of the network to match BYOL+SEM would be more convincing. [after rebuttal: addressed]


**Summary Of The Paper:**

Examines the utility of including a large number of parallel softmax operations, also called simplicial embedding (SEM), within a SSL method.  Includes theory showing improvements in generalization error bounds from using SEM, as well as extensive experimental evaluations showing benefits to classification, transfer learning, robustness to input corruption, as well as demonstrating interpretability of the SEM features (clustering by class).  The experiments also give insight on how to choose the number of simplices and the dimensionality of the simplex, as well as the temperature parameter.

**Summary Of The Review:**

This paper puts a lot of thought into presenting and analyzing a potentially high-impact addition to semi-supervised learning techniques.  The author's rebuttal addressed many concerns about the fairness of the comparisons and overclaiming. The impact of the technique (around 1-2 percent accuracy) would be interesting to a general audience but perhaps not at spotlight level.

---

> ### Author Response · Authors · 2022-11-16
> **Official answer to Reviewer Jv3d**
>
> We thank the reviewer for their feedback and questions. We are confident that answering them will strengthen the contribution of the paper.
>
> > BYOL should be trained until convergence
>
> We agree with the reviewer that we should compare BYOL + SEM with BYOL trained until convergence. We are still training our BYOL + SEM model for more epochs and will update the reviewer as soon as the training concludes. Nevertheless, we wanted to address the other reviewer’s concern as soon as possible.
>
> > Increasing the number of nonlinear layers or increasing the size of the network to match BYOL+SEM would be more convincing.
>
> We performed the experiment using the method proposed by Reviewer 4CQ8 and described in [1] Appendix F and implemented in [2]. That is, we add three blocks of [convolution, batchnorm, ReLU].  We observe that only scaling the embedding size of BYOL leads to a smaller improvement than the one achieved by SEM as shown below (also see updated Figure 3).
>
> | Emb. size -> | 128 | 130 | 256 | 650 | 1024 | 1300 | 2048 | 4096 | 6500 | 13000 | 16384| 32768|65000 | 65536|130000|131072|
> |-|-|-|-|-|-|-|-|-|-|-|-|-|-|-|-|-|
> | BYOL | 71.0 | - | 71.3 | - | 72.2 | - | 73.6 | 74.0 | - | - | 75.1 | 74.2 | -| 73.9 | -| 73.6|
> | BYOL + SEM | - | 69.1 | - | 73.4 | - | 74.7 | - | - | 76.3 | 76.7 | - | - | 77.3  | - | 77.7|-|
>
> [*] The embedding sizes have to be power of two due to a constraint to preserve the identity mapping of ResNet in the code provided in [2].
>
> > Why are some of the competing methods doing so poorly?
>
> We share the reviewer’s concern for a fair comparison. This is why, for every baseline, we take an independent implementation (when available) to which we add SEM. We do not additionally tune the SSL parameters in the model with SEM to assess the effect of the proposed method. We additionally performed the SSL comparisons on CIFAR-100 with a ResNet-50 (i.e. we solely changed the encoder architecture for the SSL method with and without SEM) and we observe similar improvement as shown in the updated Table 1 and below
>
> ||SimCLR|MOCO|BYOL|Barlow|VicReg|
> |-|-|-|-|-|-|
> |Baseline|70.5|73.24|74.23|72.03|70.8|
> |With SEM|73.24|75.78|77.48|73.28|73.30|
>
>
> > Why is BYOL+Gumbel performing so poorly?
>
> To implement BYOL+Gumbel, we replaced the softmax in SEM with PyTorch’s implementation of Gumbel softmax. Then, we performed a search over the temperature hyperparameters to find the best-performing model, which we have reported in the paper.
>
> Thanks to the reviewer’s curiosity, we further searched over the learning rates as well as a decay schedule for the temperature of the gumbels to obtain a stronger baseline. We also probed both the encoder’s output and the discrete representation. Our results are as follows:
> ||probe encoder’s representation |probe discrete representation|
> |-|-|-
> |BYOL + GST| 63.3 | 54.5 |
>
> We find that probing the encoder’s output rather than the discrete representation performs better for BYOL+GST, a setting that we have found not to perform as well with SEM. Using the encoder’s output is also the better setting for V.Q. that we have already reported in the paper.
>
> Note that our method remains better than the now stronger Gumbel softmax baseline.
>
> > The practical utility of the theoretical result is unclear.
>
> Theorem 1 is at the core of our argument about the utility of SEM for downstream classification. That is, controlling the expressivity of the SEM vectors with the temperature has an effect on the expected error of classification. We argue that this argument is of great practical relevance.

---

> > ### Author Response · Authors · 2022-11-18
> > **Update on ImageNet experiments**
> >
> > In the time we had for the rebuttal, we have been able to run a 400-epoch BYOL+SEM experiment, which already beats the 1000-epoch BYOL baseline by 0.5% (BYOL at 1000 epochs: 74.3%, BYOL+SEM at 400 epochs: 74.8%).
> >
> > Thus, we are confident that we will be able to demonstrate notable improvement over the standard BYOL setting for ImageNet when our 800-epoch training concludes, which we are currently running. We are going to update the ImageNet experimental section as well as Figure 1 when the experiments complete.
> >
> > Edit: updated the comment to indicate that BYOL numbers are for 1000 epochs, not 800. Nevertheless, our running experiments are for 800 epochs.

---

### Official Review · Reviewer_4CQ8 · 2022-10-23

**Confidence:** 4
**Correctness:** 3
**Technical Novelty And Significance:** 3
**Empirical Novelty And Significance:** 3
**Recommendation:** 8

**Clarity, Quality, Novelty And Reproducibility:**

**Clarity**: The paper is generally clear, easy to understand and enjoyable to read. My only concern in this respect is that the theoretical section is not understandable using only the main paper. Indeed one of the components of the theorem ($\psi$) is not even defined in main paper. I think that all terms in the main results should be defined in the main paper.

**Quality**: as discussed in the weakness the theory does not seem meaningful and the main claim of the paper (usefulness of SEM) is not appropriately supported.

**Novelty** the proposed method is novel

**Repredocuability** hyperparameters and code is provided.

**Strength And Weaknesses:**

**Updated during rebuttal** the authors removed some strong claims and results showing that the gains are due to the proposed method rather than only increasing dimensionality of the representation. I thus update my score 5->8 .

-------

**Strengths**
- **Paper is well written** the paper is generally easy to understand and enjoyable to read.
- **Simplicity and generality** the main strength of the transformation is that it is simple and widely applicable.
- **Significant empirical gains** the empirical results show uniformly significant gains across settings. Assuming that the baseline is indeed meaningful (see below), the results clearly indicate that the proposed method would be useful.

**Weaknesses**
- **Is the theory meaningful?** my main issue with the theory is that it might be meaningless and unrelated to SEM[^1]. Specifically, it seems that using the arguments of your theoretical section one would conclude that any transformation that shrinks the support of Z would be better. For example, let $f_{\mathrm{new}}(z)=g(z/2)$ be a function that is defined like for SEM but the transformation is halving rather than SEM (other transformations like relu also works). Then if I followed your definitions you would essentially[^2] have $\psi_{f_{new}} - \psi_{f_{base}} \leq 2V - 4V = -2V < 0 $. Using the rest of your arguments you would thus conclude that "halving representation provably leads to better performance".  You can further use a parameter $t$ instead of $2$ to say $\psi_{f_{new}} \to 0$ as $\frac{1}{t} \to 0$. For the theorem to be meaningful for SEM it should give rise to the same conclusion for any transformation just because it shrinks the support. Am I missing something?
- (addressed in rebuttal) **Are gains only due to increasing dimensionality of the representations?** my main issue with the experimental results is that it is unclear to me whether the gains are only due to the increasing dimensionality of the representation rather than to the use of SEM. In particular, I think that your experiments and previous work show that: SEM+high dim helps, SEM only does not help, and high dim only helps. It is thus currently unclear what is the gain of SEM. More specifically:
    - **Previous papers show similar gains by only increasing dimensionality** [1] argues theoretically + shows empirically that increasing dimensionality of representations helps downstream performance. There figure 7c shows a monotonic improvement in embedding size similar to BYOL+SEM in your figure 3. The line of work on dimensionality collapse, eg, [2], also suggests that higher effective dimensionality of the representation should improve performance regardless of SEM.
    - **Inadequate baseline** As just mentioned [1] finds a significant monotonic improvement when increasing dimensionality without SEM while your baseline doesn't. I think that the problem is that the baseline you are using does not actually increase the effective dimensionality of the representation. Indeed, your embedding layer is linear (linear layer + BN) and as a result, it cannot increase the dimensionality of the span of the representations (ie you are only increasing the ambient space but not the real/effective dimensionality). To bypass this issue [1] increases the dimensionality before the avg pooling layer (see Appx F1). Another standard way of increasing the dimensionality of representations of resnets which can lead to significant gains is to change the pooling layer (eg see [this code](https://github.com/facebookresearch/vissl/blob/main/configs/config/benchmark/linear_image_classification/imagenet1k/eval_resnet_8gpu_transfer_in1k_linear.yaml#L65) ). For this reason, the current figure 3 does not adequately support the hypothesis that SEM is what drives the performance up.
    - **SEM without increasing dimensionality seems to show no gains** BYOL + SEM without increasing dimension in Figure 3 seems to perform similarly to standard BYOL (table 17), thus suggesting that SEM without increasing dimensionality does not help.
 - (addressed in rebuttal) **Some over overclaiming**
    -  in the abstract, you say "we formally prove that the SEM representation leads to better generalization than normalized". This is a strong claim, which I do not think is supported given that you only show one upper bound is smaller than the other (without a discussion about their tightness). To support such a statement you could for example provide a lower bound on the generalization gap of $f_{\mathrm{base}}$ and show that it is larger than the upper bound of $f_{\mathrm{ERM}}$
    - section 4.2 you say "Figure 5 [...] allowing us to confirm two predictions made in Section 3.2: the expected generalization [...]". Your experiments do not actually test the generalization gap of the probe (as suggested by your theory) but the general test performance. One potential explanation (which would be my guess) is that the training performance of the linear probe is actually what drives the performance up (due to the increase of dimensionality) rather than the shrinking of the generalization gap. I think that showing the generalization gap is needed if you want to make any claims about the relation between your theory and experiments. Another way of testing your claims would be consider the standard "semi-supervised" setting where the downstream probe uses only 1% of ImageNet. My guess is that your representation will perform worst there (due to the high dimensionality) while your theory would suggest that it actually performs better due to the sparsity.

[^1] Note that I haven't read all appendices in detail. If the argument breaks please refer me to the exact line and I will look at it.

[^2] I removed $\delta$ for conciseness but the argument still holds with those.

[1] [Improving Self-Supervised Learning by Characterizing Idealized Representations](https://arxiv.org/abs/2209.06235)

[2] [On Feature Decorrelation in Self-Supervised Learning](https://arxiv.org/abs/2105.00470)

**Summary Of The Paper:**

The paper proposes to apply a  subcomponent-wise softmax to SSL representations (z -> partition -> softmax -> concatenate -> new z), dubbed SEM. They empirically show that this simple transformation combined with a large increase in the dimensionality of the representation yields significant improvements in standard SSL benchmarks (in and out of distribution). They also try to explain the gains by providing generalization bounds of the downstream predictors.

**Summary Of The Review:**

The paper is enjoyable to read and could be impactful if the author's main claim (usefulness of SEM) holds. My main issue with the current version is that I have doubts about: (1) whether the theory is meaningful; (2) whether the gains are due to SEM rather than increasing the dimension of the representation.

---

> ### Author Response · Authors · 2022-11-16
> **Official answer to Reviewer 4CQ8**
>
> We are pleased to read that the reviewer enjoyed reading this version of the paper. We thank the reviewer for the clear questions and are happy to address them below.
>
> > Is the theory meaningful?
>
> We agree with the reviewer’s observation that indeed $\sigma$ which downscale the $z$-space is going to decrease the corresponding $\varphi$. **We will argue however that this is not enough to have better performance**. Fitting a training set of pairs $(\frac{x}{T}, y)$ requires access to functions with larger Lipschitz constants than fitting a training set of pairs $(x, y)$. Considering the model assumption that $l_y \circ f$ belongs to a uniformly Lipschitz class of functions, there will be no model which can fit a sufficiently shrinked $z$-space to the same $y$; thus increasing the training error term in that case. Otherwise, one would have to increase R proportionally to $T$ to fit set of pairs $(\frac{x}{T}, y)$ .
>
> That being said, this does not mean that no other nontrivial transformation (like SEM) can reduce this bound. As such, the bound may be more generally applicable than to study the effect of SEM. The fact that the bound presented in Theorem 1 can be applied to other nontrivial transformation is not a weakness of the theory, but instead a strength. Moreover, this bound is an important component of the argument about the utility of SEM for downstream classification and is not present in prior works.
>
> > Are gains only due to increasing dimensionality of the representations?
>
> The reviewer shared a paper with a method to increase the dimensionality of the representation of SSL method [1]. In the proposed paper, improvement is observed for linear evaluation as the authors scale up the representation. But, the paper does not study representation’s size larger than 4096. We reproduce the experimental setup with a ResNet-50 backbone and the CIFAR-100 dataset and extend the representation’s size probed to better understand the impact of SEM using the same procedure proposed in the paper. That is, we add three blocks of [convolution, batchnorm, ReLU] before the mean-pooling and take their implementation [2] to perform the experiments. We note that the representation size are factors of 2 due to constraints in the code. We present the results below and added them in Figure 3.
>
> | Emb. size -> | 128 | 130 | 256 | 650 | 1024 | 1300 | 2048 | 4096 | 6500 | 13000 | 16384| 32768|65000 | 65536|130000|131072|
> |-|-|-|-|-|-|-|-|-|-|-|-|-|-|-|-|-|
> | BYOL | 71.0 | - | 71.3 | - | 72.2 | - | 73.6 | 74.0 | - | - | 75.1 | 74.2 | - | 73.9 | - | 73.7|
> | BYOL + SEM | - | 69.1 | - | 73.4 | - | 74.7 | - | - | 76.3 | 76.7 | - | - | 77.3 | - | 77.7|-|
>
> As hypothesized by the reviewer, we observe better performance using the proposed method than using the BYOL + Embed method initially used in the paper. However, we observe that SEM generally leads to better performance for representation size greater than 650 and is more robust to very large representations whereas increasing the representation’s size without SEM decreases after a size of 16384.
>
> > Some over-claiming
>
> We agree with the reviewer that some claims were too strong and we are thankful for pointing that out. We have softened the claims to better represent the contributions. In particular, we softened the claim “we formally prove that the SEM representation leads to better generalization than an unnormalized representation” found in the abstract and in the introduction to “we provide an upper bound and argue that using SEM leads to a better expected error than the unnormalized representation.”
>
> We want to remark that we performed the standard semi-supervised experiment with only 1% of ImageNet in Table 5 and observed a gain of 5% when using SEM over the baseline without SEM.

---

> > ### Comment · Reviewer_4CQ8 · 2022-11-17
> > **Thank you I'm raising my score.**
> >
> > Thank you for answering my main concerns (especially whether the gains were really about SEM rather than increasing dimensionality), I'm thus increasing my score.
> >
> > I am still not convinced about how meaningful the theory is (for example I would be surprised if the gains are really about decreasing the generalization gap, rather than increasing the training performance of the probe), but that can be investigated in future work.

---

> > > ### Author Response · Authors · 2022-11-18
> > > **Thank you and empirical generalization gap**
> > >
> > > We thank the reviewer for their questions and the feedback provided that allowed us to improve the clarity of the paper, especially the theoretical section.
> > >
> > > To address the remaining question, we empirically investigated the generalization gap of classifiers trained on the representation extracted from a pre-trained BYOL+SEM model with a CIFAR-100 dataset and a ResNet-50 architecture. The conclusion is that SEM improves the training accuracy of a linear probe, as conjectured by the reviewer, but also reduces the generalization gap.  In the following table, we compute the generalization gap as the difference of the training accuracy and the test accuracy of a downstream linear probe with and without SEM($\tau_d=0.1$) given the same datasets of pair (z, y) where z are the extracted unnormalized features given by the SSL model pre-trained using BYOL + SEM. We get the following generalization gap:
> > >
> > > || f_base | f_SEM |
> > > |-|-|-|
> > > | Train accuracy | 87.0  | 93.1 |
> > > | Test accuracy  | 69.5 | 77.4 |
> > > | Generalization gap | 17.5 |  16.3 |
> > >
> > >
> > > Edit to add:
> > >
> > > Our bound also holds for MLP classifiers allowing us to increase the training accuracy of f_base and test the generalization gap in a context where the training accuracy of f_base and f_SEM are closer. With a one hidden layer MLP (hidden size = 1024 and ReLU non-linearity) and the pre-training setup described above, we get the following generalization gap
> > >
> > > || f_base | f_SEM |
> > > |-|-|-|
> > > | Train accuracy | 94.4  | 94.6 |
> > > | Test accuracy  | 75.6 | 77.2 |
> > > | Generalization gap | 18.8 |  17.4 |

---

> > > > ### Comment · Reviewer_4CQ8 · 2022-11-20
> > > > **Thanks + side note**
> > > >
> > > > Interesting results, thanks for running / reporting them. I take back what I said on the theory (possibly) wrongly suggesting  gains about the probe’s generalization gap.
> > > >
> > > > Side note on the **training** accuracy: my guess for the training gains compared to increasing the dimensionality as in [1] is that SEM avoids dimensionality collapse because it applies the softmax on subcomponents of the representation. If so then SEM is probably a simple and effective solution to the problem raised in the dimensionality collapse literature. Seeing less gains for the training accuracy of MLP probes would also make sense given that those require a smaller effective dimensionality (eg section 5 in [1]).
> > > >
> > > > Good paper and I hope the authors will release some of the ImageNet pretrained models!

---

> > > > > ### Author Response · Authors · 2022-11-21
> > > > > **The reviewer is right on their interpretation of SEM and its effect on dimensionality collapse, but...**
> > > > >
> > > > > We performed an experiment where we probed the spectrum of the covariance matrix of the representation $z$, as typically done in the dimensionality collapse literature and present the result in Figure 8 of the paper. We find that SEM indeed alleviates the problem of dimensionality collapse. But, we also find that it does collapse at the particular point of 2048, which is the dimensionality of the encoder's output.
> > > > >
> > > > > > I hope the authors will release some of the ImageNet pretrained models
> > > > >
> > > > > We are happy to release the ImageNet pretrained models. We can also release the CIFAR-100 models if there is an appetite for that.

---

### Official Review · Reviewer_E9n7 · 2022-10-23

**Confidence:** 3
**Correctness:** 3
**Technical Novelty And Significance:** 3
**Empirical Novelty And Significance:** 3
**Recommendation:** 8

**Clarity, Quality, Novelty And Reproducibility:**

Clarity: Good.\
Quality: Good.\
Novelty: Good.\
Reproducibility: Good. It clearly states the base codebase and provides code in the supplementary.

**Strength And Weaknesses:**

**Strength**

This paper makes a good attempt to validate their claims, providing a theory and empirical evidence. For example, the paper provides an ablation study on the size of $L$ and $V$, a comparison with hard discretization, and an analysis of semantic coherence. The overall presentation was well-written and supported by proper evidence.

---

**Weakness**

While the paper suggests an interesting alternative to designing an embedding, my biggest concern is whether future research would really use this technique. Here are some reasons why its practicality is not convincing yet.

* **Weak baselines.** The paper shows an improvement over underfitted BYOL trained for 200 epoch (IN acc.: 70.6) instead of standard 800 epoch (IN acc.: 74.3). Since many methods converge faster but saturates eventually (or even collapse later), the paper should provide the results at 800 epoch. Also, this repo (https://github.com/HobbitLong/SupContrast) shows that SimCLR trained on CIFAR-100 shows acc. of 70.7, although they use ResNet-50, which is 5% higher than this paper. Thus, the overall baseline results are undervalued, and the gain of this method can be exaggerated. Providing the results in the SOTA setup would make the benefit of SEM more convincing.

* **CIFAR-10 results.**
The paper only demonstrates the CIFAR-100 results instead of more standard CIFAR-10. I suspect that SEM is less effective for CIFAR-10. Still, it would be informative if the paper provided CIFAR-10 results in Appendix. How can we explain this if SEM is less effective for few-class classification? Discussion for coarse vs. fine-grained classification would be informative.

* **Heavy computation.**
As the paper mentions, SEM needs an additional layer to expand the embedding overcomplete, which requires many parameters, memory, and computation. Although Table 17 in Appendix suggests an efficient version, it still uses 2~3x of parameters. Due to this, the actual baseline should be BYOL with some additional nonlinear layers instead of the vanilla BYOL.
I appreciate that the paper shows that naively adding the overcomplete layer to BYOL does not work, as shown in Figure 3. However, I think a better way to sell this method is to emphasize the scaling part -- instead of adding an overcomplete layer (which may be arguable for a fair comparison) -- achieving SOTA using very-wide models would be more impactful. For example, current self-supervised learning (SSL) methods achieve SOTA on models like ResNet-50 x4. If the paper claims that prior work does not scale for larger models such as ResNet-50 x16 but SEM can, it could be a game changer.

* **Why self-supervised learning (SSL)?**
While the paper targets SSL, the idea of SEM could also be applied to supervised learning. Why the paper's scope should be SSL? Would SEM be more effective for SSL than supervised learning?

* **Comparison with a mixture of experts (MoE).**
Conceptually, SEM may be thought of as a simple instantiation of MoE [1] or MCL [2] where each $V$-dim subvector is routing the specialized feature for each class. The bipartite in Figure 6 shows the learned features are indeed specialized to some classes. Here, the MoE technique is popularly used to train large-scale models. Since one major benefit of SEM is robust training on larger overcomplete features, it may discuss the relation to MoE and may show superiority over them.

[1] Fedus et al. A Review of Sparse Expert Models in Deep Learning. arXiv 2022.\
[2] Guzmán-rivera et al. Multiple Choice Learning: Learning to Produce Multiple Structured Outputs. NeurIPS 2012.

Typo:
- Gubel -> Gumbel in the caption of Table 2.

**Summary Of The Paper:**

This paper suggests using simplicial embedding (SEM) -- instead of the standard R^d embedding -- as the feature for self-supervised learning and downstream classification. Specifically, given $L*V$-dim feature, SEM applies softmax operation for each $V$-dim subvector and concatenates the sparsified vectors. The paper claims the superiority of SEM theoretically and empirically. First, the paper provides a new generalization bound where the complexity measure $\varphi$ is determined by the type of final representation. Here, the paper claims that the complexity measure of SEM is strictly smaller than the one of standard R^d embedding. Second, the paper demonstrates that SEM improves the classification performance of various self-supervised learning methods.

**Summary Of The Review:**

Overall, I think the paper has a clear contribution. However, I'm on the borderline since empirical evidence is not convincing enough. I'm willing to raise my score if my concerns are addressed:
- ImageNet results comparing with BYOL in 800 epoch
- CIFAR-10 results and explanation of why SEM is more effective for many-class classification
- Results on wide models such as ResNet-50 x16 instead of adding overcomplete layer
- Discussion on why SEM should be used for SSL - why not supervised learning?
- Comparison with MoE methods - would SEM be a better way to scale large models?

---

> ### Author Response · Authors · 2022-11-16
> **Official answer to Reviewer E9n7**
>
> We thank the reviewer for their comment and appreciation of the contribution. The reviewer requested several benchmarks that we ran, and are currently running, and currently observe positive results for the SSL models trained with SEM. We hope that those results will strengthen the quality of the baselines and convince the reviewer of the practicality of SEM.
>
> > Demonstrating that BYOL + SEM can outperform BYOL trained for 800 epochs
>
> We agree with the reviewer that showing that BYOL+SEM can outperform BYOL’s performance at 800 epochs would strengthen the baseline comparison. We are still training our BYOL + SEM model for more epochs and will update the reviewer as soon as the training concludes.
>
> > CIFAR-100 results with a ResNet-50 encoder
>
> We presented results with a ResNet-18 encoder because the baseline results came from an independent source which, we argue, would provide unbiased baselines. We replaced the ResNet-18 backbone for a ResNet-50 backbone and re-ran the CIFAR-100 baseline comparison for several SSL methods. We see a similar improvement from using SEM when using a ResNet-50 backbone instead of a ResNet-18 backbone as demonstrated in the following Table and in the updated Table 1.
>
> ||SimCLR|MOCO|BYOL|Barlow|VicReg|
> |-|-|-|-|-|-|
> |Baseline|70.5|73.24|74.23|72.03|70.8|
> |With SEM|73.24|75.78|77.48|73.28|73.30|
>
> > CIFAR-10 results
>
> We present CIFAR-10 results as requested by the reviewers with a ResNet-50 and the CIFAR-100 dataset on BYOL. We added the result in the Appendix as proposed by the reviewer.
>
> ||BYOL|
> |-|-|
> |Baseline| 94.2 |
> |With SEM| 95.8 |
>
> > Running with additional nonlinear layers [to match the parameter count of SEM]
>
> We performed the experiment using the method proposed by Reviewer 4CQ8 and described in [1] Appendix F and implemented in [2]. That is, we add three blocks of [convolution, batchnorm, ReLU].  We observe that only scaling the embedding size of BYOL leads to a smaller improvement than the one achieved by SEM as shown below (also see updated Figure 3).
>
> |Emb. size|128|130|256|650|1024|1300|2048|4096|6500|13000|16384|32768|65000|65536|130000|131072|
> |-|-|-|-|-|-|-|-|-|-|-|-|-|-|-|-|-|
> |BYOL|71.0|-|71.3|-|72.2|-|73.6|74.0|-|-|75.1|74.2|-|73.9|-|73.6|
> |BYOL+SEM|-|69.1|-|73.4|-|74.7|-|-|76.3|76.7|-|-|77.3|-|77.7|-|
>
> [*] The embedding sizes have to be power of two due to a constraint to preserve the identity mapping of ResNet in the code provided in [2].
>
> > Results with wide models such as ResNet-50 x 16.
>
> We appreciate the reviewer’s suggestion to study the scaling of SEM and use that as a selling point for the paper. We wish that we could make that claim, but our compute budget does not allow us to run the model scale request by the reviewer (which represents a ResNet backbone of 3.1B parameters.)
>
> We performed a scaling study on CIFAR-100 in Figure 8 and see BYOL performance degrade for wide ResNet-50 encoders. Models trained with SEM see improvement. Thus, we believe that models trained with SEM have better scaling capabilities than SSL models trained without SEM. We included the results in the appendix hoping that it would inspire researchers with more compute to run larger scale study of SEM-like approaches.
>
> > Would SEM be more effective for SSL than supervised learning?
>
> We performed an experiment where we added a SEM to the penultimate layer of a ResNet-50 classifier trained on CIFAR-100 and observed the same performance as the model trained without SEM. One main reason comes to mind to explain why SEM may be less effective in the classical supervised learning framework: The objective of supervised learning already forces the samples to cluster in a much tighter discrete bottleneck than the one induced by SEM. Thus, the SEM might not impose any structure as it is much weaker than the classification objective.
>
> > Comparing MOE and SEM
>
> We appreciate the comparison of SEM with MOE. But, we argue that they have fundamental differences and that they would be better seen as collaborating methods rather than competing ideas. The key difference is that MOE is a global routing mechanism that selects a subset of the features. If we want to compare SEM to a MOE then it would have to be a (very) local routing mechanism. I.e. SEM selects only one of V features for L independent vectors. On the other hand, MOE does not represent features and thus cannot be viewed, at least trivially, as being a version of SEM.
>
> Nevertheless, we believe that SEM and MOE might work well together. We envision an idea where the MOE would select SEM. This could reduce the memory and compute of using SEM and thus allow the scaling of SEM to bigger models. Again, we leave that exploration to researchers that have available compute to run those studies.
>
> [1] Improving Self-Supervised Learning by Characterizing Idealized Representations. https://arxiv.org/abs/2209.06235
> [2]https://github.com/YannDubs/Invariant-Self-Supervised-Learning/blob/main/issl/architectures/cnn.py#L88

---

> > ### Comment · Reviewer_E9n7 · 2022-11-16
> > **Response to the Rebuttal**
> >
> > Thank you for the careful rebuttal. My initial concerns are addressed, such as performance on fewer-class classification and supervised learning. Therefore, I increased my recommendation to accept.
> >
> > I understand that running large-scale experiments is hard in university labs. Still, updating the BYOL results to 800 epochs in the future revision will make SEM more convincing. Also, hope to see if SEM works for larger-scale models in future research.

---

> > > ### Author Response · Authors · 2022-11-18
> > > **Thank you and our update on our ImageNet experiments**
> > >
> > > We are happy that we have addressed most of your concerns. We agree that running our ImageNet experiments for 800 epochs will make the paper stronger. We are currently running these experiments, and we will update the results in the camera-ready version of our paper in the ImageNet subsection as well as Figure 1.
> > >
> > > In the time we had for the rebuttal, we have been able to run a 400-epoch BYOL+SEM experiment, which already beats the 1000-epoch BYOL baseline by 0.5% (BYOL at 1000 epochs: 74.3%, BYOL+SEM at 400 epochs: 74.8%).
> > >
> > > Thus, we are confident that we will be able to demonstrate notable improvement over the standard BYOL setting for ImageNet when our 800-epoch training concludes. We thank the reviewer for pushing us to perform this experiment which will make for a stronger camera-ready version.
> > >
> > > Edit: updated the comment to indicate that BYOL numbers are for 1000 epochs, not 800. Nevertheless, our running experiments are for 800 epochs.

---

### Official Review · Reviewer_u5uQ · 2022-10-23

**Confidence:** 4
**Correctness:** 4
**Technical Novelty And Significance:** 3
**Empirical Novelty And Significance:** 3
**Recommendation:** 8

**Clarity, Quality, Novelty And Reproducibility:**

See clarity/quality/novelty above.
Reproducibility:
The author includes very extensive descriptions of their experiment setup in the appendix and provides code in the supplemental maternal. I believe it will be sufficient to reproduce their work.

Minor comment:
Appendix D.1: the last line at page 24, "the $ of parameters" should probably be change to "the # of parameters".



**Strength And Weaknesses:**

Strength:

Clarity and quality of writing. Extensive experiments.
The paper is well written with a pretty clear introduction of background and related work. I also like how the author draws inspirations from the over-complete representation literature to motivate the proposed method. The experimental results show both quality improvement (with reasonable computation increase) and interpretability improvement.

Originality. The work seems pretty novel to me.
Practical application: Being aware of the increased memory usage from SEM, the author also studies how to efficiently reduce memory usage at inference time.

Though the findings in the proposed work is largely based on prior work in SSL, the SEM idea is pretty interesting and effective (observation from the empirical results).

Weakness:
Questions to the author(s):

1. Regarding Figure 5 (a), or effects of increasing $L$. Have you investigated when the improvement of accuracy starts to diminish if we further increases $L$ (e.g, increase $L$ to 20K, 50K)?  A larger number of basic vectors could be very helpful in defining finer granularities in some applications.

2. Could we increase the embedding dimension in BYOL to achieve the similar quality improvement as achieved by SEM?

3. Is the BYOL metric in Figure 1 a strong baseline? This is not something I am familiar with. Please clarify.

4. After SSL embedding is obtained, one usage of the SSL embedding is to perform clustering to obtain needed level of granularity and interpretability. Let's say we cluster the embeddings into 5K groups. And also on the other hand, we have the SEM embedding (when L= 5K), what would be the pros and cons of the two approaches ?

I will be willing to increase my score once the questions above are well clarified/explained.

**Summary Of The Paper:**

In self-supervised learning (SSL), a representation is learned and it is usually one multi-dimensional vector. This paper proposes an effective representation using multiple sparsified embeddings (SEM) inspired by over-complete representation. It is obtained by projecting the SSL embedding into multiple smaller dimensional vectors each through a softmax operation. The sparsity of the smaller vectors can be controlled via the temperature parameter in the softmax function. The author gives theoretical analysis on the benefit of using SEM and also conducts extensive empirical experiments. The results show the superiority of the proposed method.

**Summary Of The Review:**

The author proposes a simple yet effective component to build more interpretable and more effective representation. The experiments are solid and extensively demonstrate the effectiveness of the method. There, I think it is a good paper - "accept".

---

> ### Author Response · Authors · 2022-11-16
> **Official answer to Reviewer u5uQ**
>
> We thank the reviewer for their comment and the kind words about the writing. We are pleased to read that you appreciated this paper.  We answer the reviewer’s questions below.
>
> > Have you investigated the effect of further increasing L
>
> Using the SEM sparsification method presented in Subsection D.1. and a ResNet18 backbone allows us to scale the size of L to 20000 and 50000 as requested by the reviewer. Below, we present the result of BYOL + SEM/8 (8 blocks partition of SEM) for L in {1000, 5000, 10000, 20000, 50000} and observe that the performance saturates at L=10000 (also see Figure 7). That said, L may saturate at a different point for other applications (e.g. segmentation) as pointed out by the reviewer, but such a study is outside the scope of this paper.
>
> |  | L=1000 | L=5000 | L=10000 | L=20000 | L=50000 |
> |-|-|-|-|-|-|
> | BYOL + SEM/8 | 71.3 | 73.3  | 73.4 | 73.2 | 73.0 |
>
> > Could we increase the embedding dimension of BYOL to achieve the similar quality improvement as achieved by SEM?
>
> We performed the experiment using the method proposed by Reviewer 4CQ8 and described in [1] Appendix F and implemented in [2]. That is, we add three blocks of [convolution, batchnorm, ReLU].  We observe that only scaling the embedding size of BYOL leads to a smaller improvement than the one achieved by SEM as shown below (also see updated Figure 3).
>
> | Emb. size -> | 128 | 130 | 256 | 650 | 1024 | 1300 | 2048 | 4096 | 6500 | 13000 | 16384| 32768|65000 | 65536|130000| 131072 |
> |-|-|-|-|-|-|-|-|-|-|-|-|-|-|-|-|-|
> | BYOL | 71.0 | - | 71.3 | - | 72.2 | - | 73.6 | 74.0 | - | - | 75.1 | 74.2 | -|73.9 | -| 73.6|
> | BYOL + SEM | - | 69.1 | - | 73.4 | - | 74.7 | - | - | 76.3 | 76.7 | - | - | 77.3 |- | 77.7|-|
>
> [*] The embedding sizes have to be power of two due to a constraint to preserve the identity mapping of ResNet in the code provided in [2].
>
> > Is the BYOL metric in Figure 1 a strong baseline?
>
> BYOL is among the state-of-the-art methods for a ResNet-50 encoder and our result at 200 epochs is stronger than what is present in the literature ([3] reports 70.6 while we report 71.9.) We are training BYOL + SEM for more epochs to compare our approach with BYOL trained for 800 epochs and will update the reviewer as soon as we have the result.
>
> > What is the difference between clustering the SSL representation into 5k groups and using the SEM with L=5K?
>
> This is a great question. Conceptually, when clustering, each sample can be a member of only one of the 5K groups. On the other hand, the SEM partitioning can be viewed as a "multi-clustering" approach creating 5K soft-clusters, each of size V (e.g. V=13). That is to say that each sample is a member of 5K clusters wherein they are given a soft assignment among the 13 possible groups.
>
> To answer Reviewer u5uQ's question, the advantage of SEM in comparison to clustering as an embedding is that SEM offers a more fine-grained representation. While two samples might be similar on certain aspects (e.g. same background) they may be different on certain other aspects (e.g. different foreground). A cluster would force these two samples to be either members of the same group or of different groups. This assignment is too strong in many setups and forces the algorithm to make choices that may be suboptimal. An SEM embedding allows these two samples to be members of the same group on certain aspects (background) but on different groups on other aspects (foreground). This allows far more combinations than just using 5K clusters would allow. With flat clustering, representing as many combinations as SEM can would require allocating 13^5000 clusters, which is impractical and would result in every point lying in a unique cluster and leaving a degenerate representation without structure.
>
> The advantage of clustering is that it is a simpler representation and thus may lead to better generalization if the data can be correctly fitted into the cluster representation. That said, SEM can represent the clustering embedding (i.e. with L=1 and V=5000) and the SEM representation with large L seems better for the image datasets that we studied.
>
> Going beyond the question, SEM also has advantages during pre-training, in that it performs a soft-assignment. This procedure is more amenable to back-propagating the gradient through it during pre-training than a clustering procedure that would necessitate the use of straight-through estimators which have high variance.
>
> > Typo at page 24
>
> We thank the reviewer for catching this typo and have fixed it in the updated version.
>
> [1] Improving Self-Supervised Learning by Characterizing Idealized Representations. https://arxiv.org/abs/2209.06235
> [2]https://github.com/YannDubs/Invariant-Self-Supervised-Learning/blob/main/issl/architectures/cnn.py#L88
> [3] Exploring Simple Siamese Representation Learning. https://arxiv.org/abs/2011.10566

---

> > ### Comment · Reviewer_u5uQ · 2022-11-17
> > **Thanks for the response that clarifying my questions. Score raised.**
> >
> > The answers are clear and well supported by experiment results. Thanks.

---

> > > ### Author Response · Authors · 2022-11-18
> > > **Thank you**
> > >
> > > Thank you. We are happy to read that we answered your questions.

---

### Decision · Program_Chairs · 2023-01-20

**Decision:**

Accept: notable-top-25%

**Justification For Why Not Higher Score:**

While it's a strong paper, the overall gains are modest, and the method is can be explained quite concisely. While a short talk would be nice, I don't think there is a need for a longer one.

**Justification For Why Not Lower Score:**

I think the paper would have broad interest in the community, and there are open questions around scaling and theory which maybe others would be able to follow up on. So, I believe a spotlight could be fruitful.

**Metareview: Summary, Strengths And Weaknesses:**

This is a very nicely written paper that introduces a simple procedure, with theoretical motivation, that leads to real empirical improvement on self-supervised learning of features. Reviewers were unanimous in their high scores, with multiple reviewers raising scores after a detailed and thorough rebuttal from the authors which included additional empirical results. There's some uncertainty still around how meaningful the theory is, but reviewers are happy to leave further investigation of this to future work.

**Note From Pc:**

if the above contains the word "oral" or "spotlight" please see: "oral" presentation means -> notable-top-5% and "spotlight" means -> notable-top-25%. As stated in our emails, we are disassociating presentation type from AC recommendations